# MAPPING CONDITIONAL DISTRIBUTIONS FOR DOMAIN ADAPTATION UNDER GENERALIZED TARGET SHIFT

**Matthieu Kirchmeyer[1,2], Alain Rakotomamonjy[2,3], Emmanuel de Bézenac[1], Patrick Gallinari[1,2]**
[1]CNRS-ISIR, Sorbonne University; [2]Criteo AI Lab; [3]Université de Rouen, LITIS

## ABSTRACT

We consider the problem of unsupervised domain adaptation (UDA) between a source and a target domain under conditional and label shift a.k.a Generalized Target Shift (GeTarS). Unlike simpler UDA settings, few works have addressed this challenging problem. Recent approaches learn domain-invariant representations, yet they have practical limitations and rely on strong assumptions that may not hold in practice. In this paper, we explore a novel and general approach to align pretrained representations, which circumvents existing drawbacks. Instead of constraining representation invariance, it learns an optimal transport map, implemented as a NN, which maps source representations onto target ones. Our approach is flexible and scalable, it preserves the problem's structure and it has strong theoretical guarantees under mild assumptions. In particular, our solution is unique, matches conditional distributions across domains, recovers target proportions and explicitly controls the target generalization risk. Through an exhaustive comparison on several datasets, we challenge the state-of-the-art in GeTarS.

## 1 INTRODUCTION

Unsupervised Domain Adaptation (UDA) methods (Pan & Yang, 2010) train a classifier with labelled samples from a source domain $S$ such that its risk on an unlabelled target domain $T$ is low. This problem is ill-posed and simplifying assumptions were considered. Initial contributions focused on three settings which decompose differently the joint distribution over input and label $X \times Y$: **covariate shift** (CovS) (Shimodaira, 2000) with $p_S(Y|X) = p_T(Y|X), p_S(X) \neq p_T(X)$, **target shift** (TarS) (Zhang et al., 2013) with $p_S(Y) \neq p_T(Y)$, $p_S(X|Y) = p_T(X|Y)$ and **conditional shift** (Zhang et al., 2013) with $p_S(X|Y) \neq p_T(X|Y), p_S(Y) = p_T(Y)$. These assumptions are restrictive for real-world applications and were extended into **model shift** when $p_S(Y|X) \neq p_T(Y|X), p_S(X) \neq p_T(X)$ (Wang & Schneider, 2014; 2015) and **generalized target shift** (GeTarS) (Zhang et al., 2013) when $p_S(X|Y) \neq p_T(X|Y), p_S(Y) \neq p_T(Y)$. We consider GeTarS where a key challenge is to map the source domain onto the target one to minimize both conditional and label shifts, without using target labels. The current SOTA in Gong et al. (2016); Combes et al. (2020); Rakotomamonjy et al. (2021); Shui et al. (2021) learns domain-invariant representations and uses estimated class-ratios between domains as importance weights in the training loss. However, this approach has several limitations. First, it updates representations through adversarial alignment which is prone to well-known instabilities, especially on applications where there is no established Deep Learning architectures e.g. click-through-rate prediction, spam filtering etc. in contrast to vision. Second, to transfer representations, the domain-invariance constraint breaks the original problem structure and it was shown that this may degrade the discriminativity of target representations (Liu et al., 2019). Existing approaches that consider this issue (Xiao et al., 2019; Li et al., 2020; Chen et al., 2019) were not applied to GeTarS. Finally, generalization guarantees are derived under strong assumptions, detailed in Section 2.3, which may not hold in practice.

In this paper, we address these limitations with a new general approach, named Optimal Sample Transformation and Reweight (OSTAR), which maps pretrained representations using Optimal Transport (OT). OSTAR proposes an alternative to constraining representation invariance and performs jointly three operations: given a pretrained encoder, (i) it learns an OT map, implemented as a neural network (NN), between encoded source and target conditionals, (ii) it estimates target proportions for sample reweighting and (iii) it learns a classifier for the target domain using source labels.

OSTAR has several benefits: (i) it is flexible, scalable and preserves target discriminativity and (ii) it provides strong theoretical guarantees under mild assumptions. In summary, our contributions are:

- We propose an approach, OSTAR, to align pretrained representations under GeTarS. Without constraining representation-invariance, OSTAR jointly learns a classifier for inference on the target domain and an OT map, which maps representations of source conditionals to those of target ones under class-reweighting. OSTAR preserves target discriminativity and experimentally challenges the state-of-the-art for GeTarS.
- OSTAR implements its OT map as a NN shared across classes. Our approach is thus flexible and has native regularization biases for stability. Moreover it is scalable and generalizes beyond training samples unlike standard linear programming based OT approaches.
- OSTAR has strong theoretical guarantees under mild assumptions: its solution is unique, recovers target proportions and correctly matches source and target conditionals at the optimum. It also explicitly controls the target risk with a new Wasserstein-based bound.

Our paper is organized as follows. In Section 2, we define our problem, approach and assumptions. In Section 3, we derive theoretical results. In Section 4, we describe our implementation. We report in Section 5 experimental results and ablation studies. In Section 6, we present related work.

## 2 PROPOSED APPROACH

In this section, we successively define our problem, present our method, OSTAR and its main ideas and introduce our assumptions, used to provide theoretical guarantees for our method.

### 2.1 PROBLEM DEFINITION

Denoting $\mathcal{X}$ the input space and $\mathcal{Y} = \{1, ..., K\}$ the label space, we consider UDA between a source $S = (\mathcal{X}_S, \mathcal{Y}_S, p_S(X, Y))$ with labeled samples $\widehat{S} = \{(\mathbf{x}_S^{(i)}, y_S^{(i)})\}_{i=1}^n \in (\mathcal{X}_S \times \mathcal{Y}_S)^n$ and a target $T = (\mathcal{X}_T, \mathcal{Y}_T, p_T(X, Y))$ with unlabeled samples $\widehat{T} = \{\mathbf{x}_T^{(i)}\}_{i=1}^m \in \mathcal{X}_T^m$. We denote $\mathcal{Z} \subset \mathbb{R}^d$ a latent space and $g : \mathcal{X} \to \mathcal{Z}$ an encoder from $\mathcal{X}$ to $\mathcal{Z}$. $\mathcal{Z}_S$ and $\mathcal{Z}_T$ are the encoded source and target input domains, $\mathcal{Z}_{\widehat{S}}$ and $\mathcal{Z}_{\widehat{T}}$ the corresponding training sets and $Z$ a random variable in this space. The latent marginal probability induced by $g$ on $D \in \{S, T\}$ is defined as $\forall A \subset \mathcal{Z}, p_D^g(A) \triangleq g_\#(p_D(A))$[1]. For convenience, $\boldsymbol{p}_D^Y \in \mathbb{R}^K$ denotes the label marginal $p_D(Y)$ and $p_D(Z) \triangleq p_D^g(Z)$. In all generality, latent conditional distributions and label marginals differ across domains; this is the GeTarS assumption (Zhang et al., 2013) made in feature space $\mathcal{Z}$ rather than input space $\mathcal{X}$, as in Definition 1. GeTarS is illustrated in Figure 1a and states that representations from a given class are different across domains with different label proportions. Operating in the latent space has several practical advantages e.g. improved discriminativity and dimension reduction.

**Definition 1** (GeTarS). *GeTarS is characterized by conditional mismatch across domains i.e. $\exists j \in \{1, ..., K\}, p_S(Z|Y = j) \neq p_T(Z|Y = j)$ and label shift i.e. $\boldsymbol{p}_S^Y \neq \boldsymbol{p}_T^Y$.*

Our goal is to learn a classifier in $\mathcal{Z}$ with low target risk, using source labels. This is challenging as (i) target labels are unknown and (ii) there are two shifts to handle. We will show that this can be achieved with pretrained representations if we recover two key properties: (i) a map which matches source and target conditional distributions and (ii) target proportions to reweight samples by class-ratios and thus account for label shift. Our approach, OSTAR, achieves this objective.

### 2.2 MAPPING CONDITIONAL DISTRIBUTIONS UNDER LABEL SHIFT

We now present the components in OSTAR, their objective and the various training steps.

**Components**  The main components of OSTAR, illustrated in Figure 1b, are detailed below. These components are learned and estimated using the algorithm detailed in Section 4. They include:

- a fixed encoder $g : \mathcal{X} \to \mathcal{Z}$, defined in Section 2.1.
- a mapping $\phi : \mathcal{Z} \to \mathcal{Z}$, acting on source samples encoded by $g$.

---

[1] $f_\#\rho$ is the push-forward measure $f_\#\rho(B) = \rho\left(f^{-1}(B)\right)$, for all measurable set $B$.

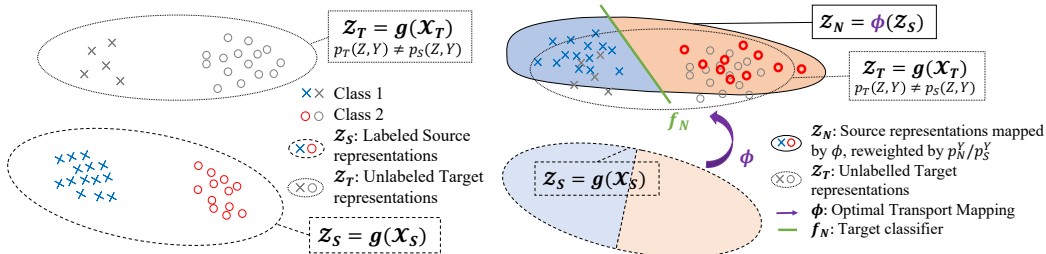

(a) UDA under `GeTarS` in latent space $\mathcal{Z}$      (b) Mapping & reweighting source samples with `OSTAR`

Figure 1: **Illustration of our approach on a 2-class UDA problem** [better viewed in color]. (a) A pretrained encoder $g$ defines a latent space $\mathcal{Z}$ with labelled source samples $\times\circ$ and unlabelled target samples $\times\circ$ under conditional and label shift (`GeTarS`). (b) We train a target classifier $f_N$ on a new domain $N$, where labelled samples $\times\circ$ are obtained by (i) mapping source samples with $\phi$ acting on conditional distributions and (ii) reweighting these samples by estimated class-ratios $\boldsymbol{p}_N^Y/\boldsymbol{p}_S^Y$. $\phi$ should match source and target conditionals and $\boldsymbol{p}_N^Y$ should estimate target proportions $\boldsymbol{p}_T^Y$.

- a label proportion vector $\boldsymbol{p}_N^Y$ on the simplex $\Delta_K$.
- a classifier $f_N : \mathcal{Z} \to \{1, \dots, K\}$ for the target domain in a hypothesis class $\mathcal{H}$ over $\mathcal{Z}$.

**Objective**   $g$ encodes source and target samples in a latent space such that it preserves rich information about the target task and such that the risk on the source domain is small. $g$ is fixed throughout training to preserve target discriminativity. $\phi$ should map encoded source conditionals in $\mathcal{Z}_S$ onto corresponding encoded target ones in $\mathcal{Z}_T$ to account for conditional shift; $\mathcal{Z}_N$ denotes the mapped space. $\boldsymbol{p}_N^Y$ should estimate the target proportions $\boldsymbol{p}_T^Y$ to account for label shift. Components $(\phi, \boldsymbol{p}_N^Y)$ define a new labelled domain in latent space $N = (\mathcal{Z}_N, \mathcal{Y}_N, p_N(Z, Y))$ through a Sample Transformation And Reweight operation of the encoded $S$ domain, as illustrated in Figure 1b. Indeed, the pushforward by $\phi$ of encoded source conditionals defines conditionals in domain $N$, $p_N^\phi(Z|Y)$:

$$\forall k, \ p_N^\phi(Z|Y = k) \triangleq \phi_\# \Big( p_S(Z|Y = k) \Big)^1$$

Then, $\boldsymbol{p}_N^Y$ weights each conditional in $N$. This yields a marginal distribution in $N$, $p_N^\phi(Z)$:

$$p_N^\phi(Z) \triangleq \sum_{k=1}^{K} \boldsymbol{p}_N^{Y=k} p_N^\phi(Z|Y = k) \tag{1}$$

Finally, classifier $f_N$ is trained on labelled samples from domain $N$. This is possible as each sample in $N$ is a projection of a labelled sample from $S$. $f_N$ can then be used for inference on $T$. We will show that it has low target risk when components $\phi$ and $\boldsymbol{p}_N^Y$ achieve their objectives detailed above.

**Training**   We train `OSTAR`'s components in two stages. First, we train $g$ along a source classifier $f_S$ from scratch by minimizing source classification loss; alternatively, $g$ can be tailored to specific problems with pretraining. Second, we jointly learn $(f_N, \phi, \boldsymbol{p}_N^Y)$ to minimize a classification loss in domain $N$ and to match target conditionals and proportions with those in domain $N$. As target conditionals and proportions are unknown, we propose a proxy problem for $(\phi, \boldsymbol{p}_N^Y)$ to match instead latent marginals $p_T(Z)$ and $p_N^\phi(Z)$ (1). We solve this proxy problem under least action principle measured by a Monge transport cost (Santambrogio, 2015), denoted $\mathcal{C}(\phi)$, as in problem (OT):

$$\min_{\phi, \boldsymbol{p}_N^Y \in \Delta_K} \mathcal{C}(\phi) \triangleq \sum_{k=1}^{K} \int_{\mathbf{z} \in \mathcal{Z}} \|\phi(\mathbf{z}) - \mathbf{z}\|_2^2 \ p_S(\mathbf{z}|Y = k) d\mathbf{z} \tag{OT}$$
$$\text{subject to } p_N^\phi(Z) = p_T(Z)$$

For any function $\phi$ e.g. parametrized by a NN, $\mathcal{C}(\phi)$ is the transport cost of encoded source conditionals by $\phi$. It uses a cost function, $c(\mathbf{x}, \mathbf{y}) = \|\mathbf{x} - \mathbf{y}\|_2^p$, where without loss of generality $p = 2$. The optimal $\mathcal{C}(\phi)$ is the sum of Wasserstein-2 distances between source conditionals and their mappings. Problem (OT) seeks to minimize $\mathcal{C}(\phi)$ under marginal matching. We provide some further

background on optimal transport in Appendix C and discuss there the differences between our OT problem (OT) and the standard Monge OT problem between $p_S(Z)$ and $p_T(Z)$.

Our OT formulation is key to the approach. First, it is at the basis of our theoretical analysis. Under Assumption 2 later defined, the optimal transport cost is the sum of the Wasserstein-2 distance between source and matched target conditionals. We can then provide conditions on these distances in Assumption 3 to formally define when the solution to problem (OT) correctly matches conditionals. Second, it allows us to learn the OT map with a NN. This NN approach: (i) generalizes beyond training samples and scales up with the number of samples unlike linear programming based OT approaches (Courty et al., 2017b) and (ii) introduces useful stability biases which make learning less prone to numerical instabilities as highlighted in de Bézenac et al. (2021); Karkar et al. (2020).

## 2.3 ASSUMPTIONS

In Section 3 we will introduce the theoretical properties of our method which offers several guarantees. As always, this requires some assumptions which are introduced below. We discuss their relations with assumptions used in related work and detail why they are less restrictive. We provide some additional discussion on their motivation and validity in Appendix D.

**Assumption 1** (Cluster Assumption on S). *$\forall k, \boldsymbol{p}_S^{Y=k} > 0$ and there is a partition of the source domain $\mathcal{Z}_S$ such that $\mathcal{Z}_S = \cup_{k=1}^K \mathcal{Z}_S^{(k)}$ and $\forall k \, p_S(Z \in \mathcal{Z}_S^{(k)} | Y = k) = 1$.*

Assumption 1, inspired from Chapelle et al. (2010), states that source representations with the same label are within the same cluster. It helps guarantee that only one map is required to match source and target conditionals. Assumption 1 is for instance satisfied when the classification loss on the source domain is zero which corresponds to the training criterion of our encoder $g$. Interestingly other approaches e.g. Combes et al. (2020); Rakotomamonjy et al. (2021) assume that it also holds for target representations; this is harder to induce as target labels are unknown.

**Assumption 2** (Conditional matching). *A mapping $\phi$ solution to our matching problem in problem (OT) maps a source conditional to a target one i.e. $\forall k \, \exists j \, \phi_\#(p_S(Z|Y=k)) = p_T(Z|Y=j)$.*

Assumption 2 guarantees that mass of a source conditional will be entirely transferred to a target conditional; $\phi$ solution to problem (OT) will then perform optimal assignment between conditionals. It is less restrictive than alternatives: the *ACons* assumption in Zhang et al. (2013); Gong et al. (2016) states the existence of a map matching the right conditional pairs i.e. $j = k$ in Assumption 2, while the *GLS* assumption of Combes et al. (2020) imposes that $p_S(Z|Y) = p_T(Z|Y)$. *GLS* is thus included in Assumption 2 when $j = k$ and $\phi = \mathrm{Id}$ and is more restrictive.

**Assumption 3** (Cyclical monotonicity between S and T). *For all $K$ elements permutation $\sigma$, conditional probabilities in the source and target domains satisfy $\sum_{k=1}^K \mathcal{W}_2(p_S(Z|Y=k), p_T(Z|Y=k)) \leq \sum_{k=1}^K \mathcal{W}_2(p_S(Z|Y=k), p_T(Z|Y=\sigma(k))$ with $\mathcal{W}_2$, the Wasserstein-2 distance.*

Assumption 3, introduced in Rakotomamonjy et al. (2021), formally defines settings where conditionals are guaranteed to be correctly matched with an optimal assignment in the latent space under Assumption 2. One sufficient condition for yielding this assumption is when $\forall k, j, \, \mathcal{W}_2(p_S(Z|Y=k), p_T(Z|Y=k)) \leq \mathcal{W}_2(p_S(Z|Y=k), p_T(Z|Y=j))$. This last condition is typically achieved when conditionals between source and target of the same class are "sufficiently near" to each other.

**Assumption 4** (Conditional linear independence on T). *$\{p_T(Z|Y=k)\}_{k=1}^K$ are linearly independent implying $\forall k, \nexists \, \alpha \in \left\{ \Delta_K | \, p_T(Z|Y=k) = \sum_{j=1, j \neq k}^K \alpha_j p_T(Z|Y=j) \right\}$.*

Assumption 4 is standard and seen in `TarS` to guarantee correctly estimating target proportions (Redko et al., 2019; Garg et al., 2020). It discards pathological cases like when $\exists (i, j, k) \, \exists (a, b) \, s.t. \, a + b = 1, p_T(Z|Y=i) = a \times p_T(Z|Y=j) + b \times p_T(Z|Y=k)$. It is milder than its alternative *A2Cons* in Zhang et al. (2013); Gong et al. (2016) which states linear independence of linear combinations of source and target conditionals, in $\mathcal{X}$ respectively $\mathcal{Z}$.

## 3 THEORETICAL RESULTS

We present our theoretical results, with proofs in Appendix E, for `OSTAR` under our mild assumptions in Section 2.3. We first analyze in Section 3.1 the properties of the solution to our problem in

(OT). Then, we show in Section 3.2 that given the learned components $g$, $\phi$ and $\boldsymbol{p}_N^Y$, the target generalization error of a classifier in the hypothesis space $\mathcal{H}$ can be upper-bounded by different terms including its risk on domain $N$ and the Wasserstein-1 distance between marginals in $N$ and $T$.

## 3.1 PROPERTIES OF THE SOLUTION TO THE OT ALIGNMENT PROBLEM

**Proposition 1** (Unicity and match). *For any encoder $g$ which defines $\mathcal{Z}$ satisfying Assumption 1, 2, 3, 4, there is a unique solution $(\phi, \boldsymbol{p}_N^Y)$ to* (OT) *and $\phi_\#(p_S(Z|Y)) = p_T(Z|Y)$ and $\boldsymbol{p}_N^Y = \boldsymbol{p}_T^Y$.*

Given an encoder $g$ satisfying our assumptions, Proposition 1 shows two strong results. First, the solution to problem (OT) exists and is unique. Second, we prove that this solution defines a domain $N$, via the sample transformation and reweight operation defined in (1), where encoded conditionals and label proportions are equal to target ones. For comparison, Combes et al. (2020); Shui et al. (2021) recover target proportions only under *GLS* i.e. when conditionals are already matched, while we match both conditional and label proportions under the more general `GeTarS`.

## 3.2 CONTROLLED TARGET GENERALIZATION RISK

We now characterize the generalization properties of a classifier $f_N$ trained on this domain $N$ with a new general upper-bound. First, we introduce some notations; given an encoder $g$ onto a latent space $\mathcal{Z}$, we define the risk of a classifier $f$ as $\epsilon_D^g(f) \triangleq \mathbb{E}_{\mathbf{z} \sim p_D(Z,Y)}[f(\mathbf{z}) \neq y]$ with $D \in \{S, T, N\}$.

**Theorem 1** (Target risk upper-bound). *Given a fixed encoder $g$ defining a latent space $\mathcal{Z}$, two domains $N$ and $T$ satisfying cyclical monotonicity in $\mathcal{Z}$, assuming that we have $\forall k, \boldsymbol{p}_N^{Y=k} > 0$, then $\forall f_N \in \mathcal{H}$ where $\mathcal{H}$ is a set of $M$-Lipschitz continuous functions over $\mathcal{Z}$, we have*

$$\epsilon_T^g(f_N) \leq \underbrace{\epsilon_N^g(f_N)}_{\text{Classification } (C)} + \frac{2M}{\min_{k=1}^K \boldsymbol{p}_N^{Y=k}} \underbrace{\mathcal{W}_1\Big(p_N(Z), p_T(Z)\Big)}_{\text{Alignment } (A)} +$$

$$2M(1 + \frac{1}{\min_{k=1}^K \boldsymbol{p}_N^{Y=k}}) \underbrace{\mathcal{W}_1\Big(\sum_{k=1}^K \boldsymbol{p}_T^{Y=k} p_T(Z|Y=k), \sum_{k=1}^K \boldsymbol{p}_N^{Y=k} p_T(Z|Y=k)\Big)}_{\text{Label } (L)} \qquad (2)$$

We first analyze our upper-bound in (2). The target risk of $f_N$ is controlled by three main terms: the first (C) is the risk of $f_N$ on domain $N$, the second (A) is the Wasserstein-1 distance between latent marginals of domain $N$ and $T$, the third term (L) measures a divergence between label distributions using, as a proxy, two ad-hoc marginal distributions. There are two other terms in (2): first, a Lipschitz-constant $M$ that can be made small by implementing $f_N$ as a NN with piece-wise linear activations and regularized weights; second the minimum of $\boldsymbol{p}_N^Y$, which says that generalization is harder when a target class is less represented. We learn `OSTAR`'s components to minimize the r.h.s of (2). Terms (C) and (A) can be explicitly minimized, respectively by training a classifier $f_N$ on domain $N$ and by learning $(\phi, \boldsymbol{p}_N^Y)$ to match marginals of domains $N$ and $T$. Term (L) is not computable, yet its minimization is naturally handled by `OSTAR`. Indeed, term (L) is minimal when $\boldsymbol{p}_N^Y = \boldsymbol{p}_T^Y$ under Assumption 4 per Redko et al. (2019). With `OSTAR`, this sufficient condition is guaranteed in Proposition 1 by the solution to problem (OT) under our mild assumptions in Section 2.3. This solution defines a domain $N$ for which $\boldsymbol{p}_N^Y = \boldsymbol{p}_T^Y$ and term (A) equals zero.

We now detail the originality of this result over existing Wasserstein-based generalization bounds. First, our upper-bound is general and can be explicitly minimized even when target labels are unknown unlike Shui et al. (2021) or Combes et al. (2020) which require knowledge of $\boldsymbol{p}_T^Y$ or its perfect estimation. Combes et al. (2020) claims that correct estimation of $\boldsymbol{p}_T^Y$ i.e. (L) close to zero, is achieved under *GLS* i.e. $p_S(Z|Y) = p_T(Z|Y)$. However, *GLS* is hardly guaranteed when the latent space is learned: a sufficient condition in Combes et al. (2020) requires knowing $\boldsymbol{p}_T^Y$, which is unrealistic in UDA. Second, our bound is simpler than the one in Rakotomamonjy et al. (2021): in particular, it removes several redundant terms which are unmeasurable due to unknown target labels.

## 4 IMPLEMENTATION

Our solution, detailed below, minimizes the generalization bound in (2). It implements $f_N, g$ with NNs and $\phi$ with a residual NN. Our solution jointly solves (i) a classification problem to account for term (C) in (2) and (ii) an alignment problem on pretrained representations (OT) to account for terms (A) and (L) in (2). Our pseudo-code and runtime / complexity analysis are presented in Appendix F.

**Encoder initialization**   Prior to learning $\phi, \boldsymbol{p}_N^Y, f_N$, we first learn the encoder $g$ jointly with a source classifier $f_S$ to yield a zero source classification loss via (3). With $\mathcal{L}_{ce}$ the cross-entropy loss,

$$\min_{f_S, g} \mathcal{L}_c^g(f_S, S) \triangleq \min_{f_S, g} \frac{1}{n} \sum_{i=1}^{n} \mathcal{L}_{ce}(f_S \circ g(\mathbf{x_S^{(i)}}), y_S^{(i)}) \tag{3}$$

$g$ is then fixed to preserve the original problem structure. This initialization step helps enforce Assumption 1 and could alternatively be replaced by directly using a pretrained encoder if available.

**Joint alignment and classification**   We then solve alternatively (i) a classification problem on domain $N$ w.r.t. $f_N$ to account for term (C) in (2) and (ii) the problem (OT) w.r.t. $(\phi, \boldsymbol{p}_N^Y)$ to account for term (A). Term (L) in (2) is handled by matching $\boldsymbol{p}_N^Y$ and $\boldsymbol{p}_T^Y$ through the minimization of problem (OT). $\boldsymbol{p}_N^Y$ is estimated with the confusion-based approach in Lipton et al. (2018), while $\phi$ minimizes the Lagrangian relaxation, with hyperparameter $\lambda_{OT}$, of the constrained optimization problem (OT). The equality constraint in (OT) is measured through a Wasserstein distance as our bound in (2) is tailored to this distance, however, we are not restricted by this choice as other discrepancy measures are also possible. The objective based on (2) corresponds to the following optimization problem:

$$\min_{\phi, f_N} \mathcal{L}_c^g(f_N, N) + \lambda_{OT}\ \mathcal{L}_{OT}^g(\phi) + \mathcal{L}_{wd}^g(\phi, \boldsymbol{p}_N^Y)$$

$$\text{(CAL)}$$

$$\text{subject to } \boldsymbol{p}_N^Y = \underset{\mathbf{p} \geq 0, \mathbf{p} \in \Delta_K}{\arg\min} \frac{1}{2} \|\hat{\boldsymbol{p}}_T^Y - \hat{\mathbf{C}} \frac{\mathbf{p}}{\boldsymbol{p}_S^Y}\|_2^2 \qquad \text{[Label proportion estimation]}$$

$$\text{where } \mathcal{L}_c^g(f_N, N) \triangleq \frac{1}{n} \sum_{i=1}^{n} \frac{\boldsymbol{p}_N^{y_S^{(i)}}}{\boldsymbol{p}_S^{y_S^{(i)}}} \mathcal{L}_{ce}(f_N \circ \phi \circ g(\mathbf{x_S^{(i)}}), y_S^{(i)}) \qquad \text{[Classification loss in } \mathcal{Z}_{\hat{N}}\text{]} \tag{4}$$

$$\text{and } \mathcal{L}_{OT}^g(\phi) \triangleq \sum_{k=1}^{K} \frac{1}{\#\{y_S^{(i)} = k\}_{i \in [\![1,n]\!]}} \sum_{y_S^{(i)} = k, i \in [\![1,n]\!]} \|\phi(\mathbf{z_S^{(i)}}) - \mathbf{z_S^{(i)}}\|_2^2 \quad \text{[Objective function of (OT)]} \tag{5}$$

$$\text{and } \mathcal{L}_{wd}^g(\phi, \boldsymbol{p}_N^Y) \triangleq \sup_{\|v\|_L \leq 1} \frac{1}{n} \sum_{i=1}^{n} \frac{\boldsymbol{p}_N^{y_S^{(i)}}}{\boldsymbol{p}_S^{y_S^{(i)}}} v \circ \phi(\mathbf{z_S^{(i)}}) - \frac{1}{m} \sum_{j=1}^{m} v(\mathbf{z_T^{(j)}}) \quad \text{[Relaxed equality constraint in (OT)]} \tag{6}$$

$\hat{\mathbf{C}}$ is the confusion matrix of $f_N$ on domain $N$, $\hat{\boldsymbol{p}}_T^Y \in \Delta_K$ is the target label proportions estimated with $f_N$. $\mathcal{L}_c^g(f_N, N)$ (4) is the classification loss of $f_N$ on $\mathcal{Z}_{\hat{N}}$, derived in Appendix E, which minimizes term (C) in (2). Note that samples in domain $N$ are obtained by mapping source samples with $\phi$; they are reweighted to account for label shift. $\mathcal{L}_{OT}^g$ (5) defines the transport cost of $\phi$ on $\mathcal{Z}_{\hat{S}}$. Implementing $\phi$ with a ResNet performed better than standard MLPs, thus we minimize in practise the dynamical transport cost, better tailored to residual maps and used in de Bézenac et al. (2021); Karkar et al. (2020). $\mathcal{L}_{wd}^g$ (6) is the empirical form of the dual Wasserstein-1 distance between $p_N^\phi(Z)$ and $p_T(Z)$ and seeks at enforcing the equality constraint in problem (OT) i.e. minimizing term (A). OSTAR's assumptions also guarantee that term (L) is small at the optimum.

**Improve target discriminativity**   OSTAR solves a transfer problem with pretrained representations, but cannot change their discriminativity in the target. This may harm performance when the encoder $g$ is not well suited for the target. Domain-invariant methods are less prone to this issue as they update target representations. In our upper-bound in (2), target discriminativity is assessed by the value of term (C) at optimum of the alignment problem (OT). This value depends on $g$ and may be high since $g$ has been trained with only source labels. Nevertheless, it can be reduced by updating $g$ using target outputs such that class separability for target representations is better enforced. To achieve this goal, we propose an extension of (CAL) using Information Maximization

(IM), not considered in existing domain-invariant `GeTarS` methods. IM was originally introduced for source-free adaptation without alignment (Liang et al., 2020). In this context, target predictions are prone to errors and there is no principled way to mitigate this problem. In our case, we have access to source samples and `OSTAR` minimizes an upper-bound to its target error, which avoids performance degradation. IM refines the decision boundaries of $f_N$ with two terms on target samples. $\mathcal{L}_{ent}^g(f_N, T)$ (7) is the conditional entropy of $f_N$ which favors low-density separation between classes. Denoting $\delta_k(\cdot)$ the $k$-th component of the softmax function,

$$\mathcal{L}_{ent}^g(f_N, T) = \sum_{i=1}^m \sum_{k=1}^K \delta_k(f_N \circ g(\mathbf{x_T^{(i)}})) \log(\delta_k(f_N \circ g(\mathbf{x_T^{(i)}}))) \tag{7}$$

$\mathcal{L}_{div}^g(f_N, T)$ (8) promotes diversity by regularizing the average output of $f_N \circ g$ to be uniform on $T$. It avoids predictions from collapsing to the same class thus softens the effect of conditional entropy.

$$\mathcal{L}_{div}^g(f_N, T) = \sum_{k=1}^K \hat{p}_k \log \hat{p}_k = D_{KL}(\hat{p}, \frac{1}{K}\mathbf{1}_K) - \log K; \hat{p} = \mathbb{E}_{\mathbf{x_T} \in \mathcal{X}_T}[\delta(f_N \circ g(\mathbf{x_T}))] \tag{8}$$

Our variant introduces two additional steps in the learning process. First, the latent space is fixed and we optimize $f_N$ with IM in (SS). Then, we optimize the representations in (SSg) while avoiding modifying source representations by including $\mathcal{L}_c^g(f_S, S)$ (3) with a fixed source classifier $f_S$.

$$\min_{f_N} \mathcal{L}_c^g(f_N, N) + \mathcal{L}_{ent}^g(f_N, T) + \mathcal{L}_{div}^g(f_N, T) \tag{SS}$$

$$\min_{f_N, g} \mathcal{L}_c^g(f_N, N) + \mathcal{L}_{ent}^g(f_N, T) + \mathcal{L}_{div}^g(f_N, T) + \mathcal{L}_c^g(f_S, S) \tag{SSg}$$

## 5 EXPERIMENTAL RESULTS

We now present our experimental results on several UDA problems under `GeTarS` and show that `OSTAR` outperforms recent SOTA baselines. The `GeTarS` assumption is particularly relevant on our datasets as encoded conditional distributions do not initially match as seen in Appendix Figure 3.

**Setting** We consider: (i) an academic benchmark `Digits` with two adaptation problems between USPS (Hull, 1994) and MNIST (LeCun et al., 1998), (ii) a synthetic to real images adaptation benchmark `VisDA12` (Peng et al., 2017) and (iii) two object categorizations problems `Office31` (Saenko et al., 2010), `OfficeHome` (Venkateswara et al., 2017) with respectively six and twelve adaptation problems. The original datasets have fairly balanced classes, thus source and target label distributions are similar. This is why we subsample our datasets to make label proportions dissimilar across domains as detailed in Appendix Table 8. For `Digits`, we subsample the target domain and investigate three settings - balanced, mild and high as Rakotomamonjy et al. (2021). For other datasets, we modify the source domain by considering 30% of the samples coming from the first half of their classes as Combes et al. (2020). We report a `Source` model, trained using only source samples without adaptation, and various UDA methods: two `CovS` methods and three recent SOTA `GeTarS` models. Chosen UDA methods learn invariant representations with reweighting in `GeTarS` models or without reweighting in other baselines. The two `CovS` baselines are `DANN` (Ganin et al., 2016) which approaches $\mathcal{H}$-divergence and `WD`$_{\beta=0}$ (Shen et al., 2018) which computes Wasserstein distance. The `GeTarS` baselines are Wu et al. (2019); Rakotomamonjy et al. (2021); Combes et al. (2020). We use Wasserstein distance to learn invariant representations such that only the strategy to account for label shift differs. Wu et al. (2019), denoted `WD`$_\beta$, performs assymetric alignment with parameter $\beta$, for which we test different values ($\beta \in \{1, 2\}$). `MARSc`, `MARSg` (Rakotomamonjy et al., 2021) and `IW-WD` (Combes et al., 2020) estimate target proportions respectively with optimal assignment or with the estimator in Lipton et al. (2018) also used in `OSTAR`. We report `DI-Oracle`, an oracle which learns invariant representations with Wasserstein distance and makes use of true class-ratios. Finally, we report `OSTAR` with and without IM. All baselines are reimplemented for a fair comparison with the same NN architectures detailed in Appendix G.

**Results** In Table 1, we report, over 10 runs, mean and standard deviations for balanced accuracy, i.e. average recall on each class. This metric is suited for imbalanced problems (Brodersen et al., 2010). For better visibility, results are aggregated over all imbalance settings of a dataset (line "subsampled"). In the Appendix, full results are reported in Table 3 along the $\ell_1$ target proportion

Table 1: Balanced accuracy ($\uparrow$) over 10 runs. The best performing model is indicated in **bold**. Results are aggregated over all imbalance scenarios and adaptation problems within a same dataset.

| Setting | Source | DANN | $WD_{\beta=0}$ | $WD_{\beta=1}$ | $WD_{\beta=2}$ | MARSg | MARSc | IW-WD | OSTAR | OSTAR+IM | DI-Oracle |
|---|---|---|---|---|---|---|---|---|---|---|---|
| Digits | | | | | | | | | | | |
| balanced | $74.98 \pm 3.8$ | $90.81 \pm 1.3$ | $92.63 \pm 1.0$ | $82.80 \pm 4.7$ | $76.07 \pm 7.1$ | $92.18 \pm 2.2$ | $94.91 \pm 1.4$ | $95.89 \pm 0.5$ | $91.66 \pm 0.9$ | $\mathbf{97.51 \pm 0.3}$ | $96.90 \pm 0.2$ |
| subsampled | $75.05 \pm 3.1$ | $89.91 \pm 1.5$ | $89.45 \pm 1.0$ | $81.56 \pm 4.8$ | $77.77 \pm 6.5$ | $91.87 \pm 2.0$ | $93.75 \pm 1.4$ | $93.22 \pm 1.1$ | $88.39 \pm 1.5$ | $\mathbf{96.69 \pm 0.7}$ | $96.43 \pm 0.3$ |
| VisDA12 | | | | | | | | | | | |
| original | $48.63 \pm 1.0$ | $53.72 \pm 0.9$ | $57.40 \pm 1.1$ | $47.56 \pm 0.8$ | $36.21 \pm 1.8$ | $55.62 \pm 1.6$ | $55.33 \pm 0.8$ | $51.88 \pm 1.6$ | $50.37 \pm 0.6$ | $\mathbf{59.24 \pm 0.5}$ | $57.61 \pm 0.3$ |
| subsampled | $42.46 \pm 1.4$ | $47.57 \pm 0.9$ | $47.32 \pm 1.4$ | $41.48 \pm 1.6$ | $31.83 \pm 3.0$ | $55.00 \pm 1.9$ | $51.86 \pm 2.0$ | $50.65 \pm 1.5$ | $49.05 \pm 0.9$ | $\mathbf{58.84 \pm 1.0}$ | $55.77 \pm 1.1$ |
| Office31 | | | | | | | | | | | |
| subsampled | $74.50 \pm 0.5$ | $76.13 \pm 0.3$ | $76.24 \pm 0.3$ | $74.23 \pm 0.5$ | $72.40 \pm 1.8$ | $80.20 \pm 0.4$ | $80.00 \pm 0.5$ | $77.28 \pm 0.4$ | $76.19 \pm 0.8$ | $\mathbf{82.61 \pm 0.4}$ | $81.07 \pm 0.3$ |
| OfficeHome | | | | | | | | | | | |
| subsampled | $50.56 \pm 2.8$ | $50.87 \pm 1.05$ | $53.47 \pm 0.7$ | $52.24 \pm 1.1$ | $49.48 \pm 1.3$ | $56.60 \pm 0.4$ | $56.22 \pm 0.6$ | $54.87 \pm 0.4$ | $54.64 \pm 0.7$ | $\mathbf{59.51 \pm 0.4}$ | $57.97 \pm 0.3$ |

estimation error for `GeTarS` baselines and `OSTAR+IM` in Figure 4. First, we note that low estimation error of $\boldsymbol{p}_T^Y$ is correlated to high accuracy for all models and that `DI-Oracle` upper-bounds domain-invariant approaches. This shows the importance of correctly estimating $\boldsymbol{p}_T^Y$. Second, we note that `OSTAR` improves `Source` (column 2) and is competitive w.r.t. `IW-WD` (column 9) on `VisDA`, `Office31` and `OfficeHome` and `MARSg` (column 7) on `Digits` although it keeps target representations fixed unlike these two methods. `OSTAR+IM` (column 11) is less prone to initial target discriminativity and clearly outperforms or equals the baselines on both balanced accuracy and proportion estimation. It even improves `DI-Oracle` (column 12) for balanced accuracy despite not using true class-ratios. This (i) shows the benefits of not constraining domain-invariance which may degrade target discriminativity especially under label estimation errors; (ii) validates our theoretical results which show that `OSTAR` controls the target risk and recovers target proportions. We visualize how `OSTAR` aligns source and target representations in Appendix Figure 3.

**Ablation studies** We perform two studies. We first measure the **role of IM in our model**. In Appendix Table 4, we show the contribution of (SS) (column 3) and (SSg) (column 4) to (CAL) (column 2). We also evaluate the effect of IM on our baselines in Appendix Table 5, even if this is not part of their original work. IM improves performance on `VisDA` and `Office` and degrades it on `Digits`. The performance remains below the ones of `OSTAR+IM`. Finally, we evaluate the impact of IM on our upper-bound in (2) in Appendix Table 6. We assume that target conditionals are known to compute term (L). We observe that term (C), related to target discriminativity, and the alignment terms (A) and (L) are reduced. This explains the improvements due to IM and shows empirically that IM helps better optimize the three functions $\phi, g, f_N$ in our upper-bound. The second study in Table 2 measures the **effect of the OT transport cost** in our objective (CAL). Proposition 1 showed that OT guarantees recovering target proportions and matching conditionals. We consider MNIST$\rightarrow$USPS under various shifts (Line 1) and initialization gains i.e. the standard deviation of the weights of the NN (Line 2). On line 1, we note that $\lambda_{OT} \neq 0$ in problem (CAL) improves balanced accuracy (left) and $\ell_1$ estimation error (middle) over $\lambda_{OT} = 0$ over all shifts, especially high ones. This improvement is correlated with lower mean and standard deviation of the normalized transport cost per sample (right). We observe the same trends when changing initialization gains in Line 2. This confirms our theoretical results and shows the advantages of OT regularization biases for performance and stability. In Appendix Table 7, we test additional values of $\lambda_{OT}$ and see that `OSTAR` recovers the `Source` model under high $\lambda_{OT}$. Indeed, the latter constrains $\phi \approx \text{Id}$.

## 6 RELATED WORK

**UDA** Existing approaches train a classifier using source labels while handling distribution shifts. We review two main approaches: the first learns invariant representations and the second learns a map between fixed samples from source and target domains. While domain-invariant approaches are SOTA, mapping-based approaches avoid structure issues posed by the invariance constraint which may degrade target discriminativity (Liu et al., 2019; Chen et al., 2019). For `CovS`, domain-invariant methods directly match marginal distributions in latent space (Ganin et al., 2016; Shen et al., 2018; Long et al., 2015), while mapping-based approaches learn a mapping in input space (Courty et al., 2017b; Hoffman et al., 2018). For `TarS` and `GeTarS`, label shift requires estimating $\boldsymbol{p}_T^Y$ to reweight source samples by estimated class-ratios (Zhao et al., 2019). An alternative is to use a fixed weight

Table 2: Effect of OT on balanced accuracy $\uparrow$ (left), $\ell_1$ estimation error $\downarrow$ (middle) and normalized transport cost $\mathcal{L}_{OT}^g/n \downarrow$ (right) for MNIST$\rightarrow$USPS. The best model for accuracy is in **bold** with a "$\star$". We consider varying shifts (Line 1) and initialization gains (stdev of the NN's weights) (Line2)

| MNIST$\rightarrow$USPS - initialization gain 0.02 | | |
|---|---|---|
| Shift | $\lambda_{OT} = 0$ | $\lambda_{OT} = 10^{-2}$ |
| balanced | **94.92 ± 0.6** | **95.12 ± 0.6** |
| mild | 88.28 ± 1.5 | **91.77 ± 1.2** |
| high | 85.24 ± 1.6 | **88.55 ± 1.1** |

| MNIST$\rightarrow$USPS - high imbalance | | |
|---|---|---|
| Init Gain | $\lambda_{OT} = 0$ | $\lambda_{OT} = 10^{-2}$ |
| 0.02 | 85.24 ± 1.6 | **88.55 ± 1.1** |
| 0.1 | 84.62 ± 2.3 | **88.41 ± 1.3** |
| 0.3 | 83.11 ± 2.4 | **89.41 ± 1.6** |

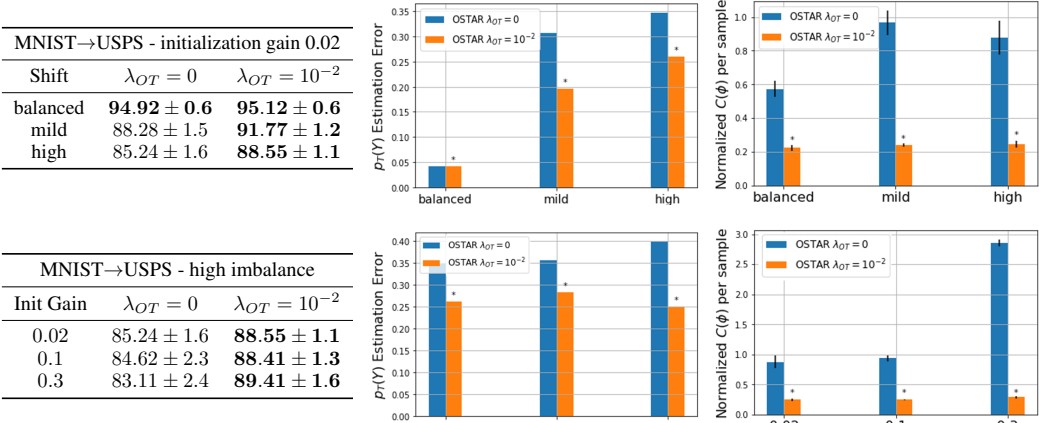

(Wu et al., 2019). When conditionals are unchanged i.e. TarS, $\boldsymbol{p}_T^Y$ can be recovered without needs for alignment (Lipton et al., 2018; Redko et al., 2019). Under GeTarS, there is the additional difficulty of matching conditionals. The SOTA methods for GeTarS are domain-invariant with sample reweighting (Rakotomamonjy et al., 2021; Combes et al., 2020; Gong et al., 2016). An earlier mapping-based approach was proposed in Zhang et al. (2013) to align conditionals, also under reweighting. Yet, it is not flexible, operates on high dimensional input spaces and does not scale up to large label spaces as it considers a linear map for each pair of conditionals. Estimators used in GeTarS are confusion-based in Combes et al. (2020); Shui et al. (2021); derived from optimal assignment in Rakotomamonjy et al. (2021) or from the minimization of a reweighted MMD between marginals in Zhang et al. (2013); Gong et al. (2016). OSTAR is a new mapping-based approach for GeTarS with improvements over these approaches as detailed in this paper. The improvements are both practical (flexibility, scalability, stability) and theoretical. CTC (Gong et al., 2016) also operates on representations, yet OSTAR simplifies learning by clearly separating alignment and encoding operations, intertwined in CTC as both encoder and maps are trained to align.

**OT for UDA**    A standard approach is to compute a transport plan between empirical distributions with linear programming (LP). LP is used in CovS to align joint distributions (Courty et al., 2017a; Damodaran et al., 2018) or to compute a barycentric mapping in input space (Courty et al., 2017b); and in TarS to estimate target proportions through minimization of a reweighted Wasserstein distance between marginals (Redko et al., 2019). An alternative to LP is to learn invariant representation by minimizing a dual Wasserstein-1 distance with adversarial training as Shen et al. (2018) in CovS or, more recently, Rakotomamonjy et al. (2021); Shui et al. (2021) for GeTarS. Rakotomamonjy et al. (2021) formulates two separate OT problems for class-ratio estimation and conditional alignment and Shui et al. (2021) is the OT extension of Combes et al. (2020) to multi-source UDA. OSTAR is the first OT GeTarS approach that does not learn invariant representations. It has several benefits: (i) representation are kept separate, thus target discriminativity is not deteriorated, (ii) a single OT problem is solved unlike Rakotomamonjy et al. (2021), (iii) the OT map is implemented with a NN which generalizes beyond training samples and scales up with the number of samples unlike barycentric OT maps; (iv) it improves efficiency of matching by encoding samples, thereby improving discriminativity and reducing dimensionality, unlike Courty et al. (2017b).

## 7    CONCLUSION

We introduced OSTAR, a new general approach to align pretrained representations under GeTarS, which does not constrain representation invariance. OSTAR learns a flexible and scalable map between conditional distributions in the latent space. This map, implemented as a ResNet, solves an OT matching problem with native regularization biases. Our approach provides strong generalization guarantees under mild assumptions as it explicitly minimizes a new upper-bound to the target risk. Experimentally, it challenges recent invariant GeTarS methods on several UDA benchmarks.

**Acknowledgement** We acknowledge financial support from DL4CLIM ANR-19-CHIA- 0018-01, RAIMO ANR-20-CHIA-0021-01, OATMIL ANR-17-CE23-0012 and LEAUDS ANR-18-CE23-0020 Projects of the French National Research Agency (ANR).

**Reproducibility Statement** We provide explanations of our assumptions in Section 2.3 and provide our proofs in Appendix E. The datasets used in our experiments are public benchmarks and we provide a complete description of the data processing steps and NN architectures in Appendix G. Our source code is available at `https://github.com/mkirchmeyer/ostar`.

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

# A    ADDITIONAL VISUALISATION

We visualize how `OSTAR` maps source representations onto target ones under `GeTarS`. We note that `OSTAR` (i) maps source conditionals to target ones (blue and green points are matched c.f. left), (ii) matches conditionals of the same class ("v" and "o" of the same colour are matched c.f. right).

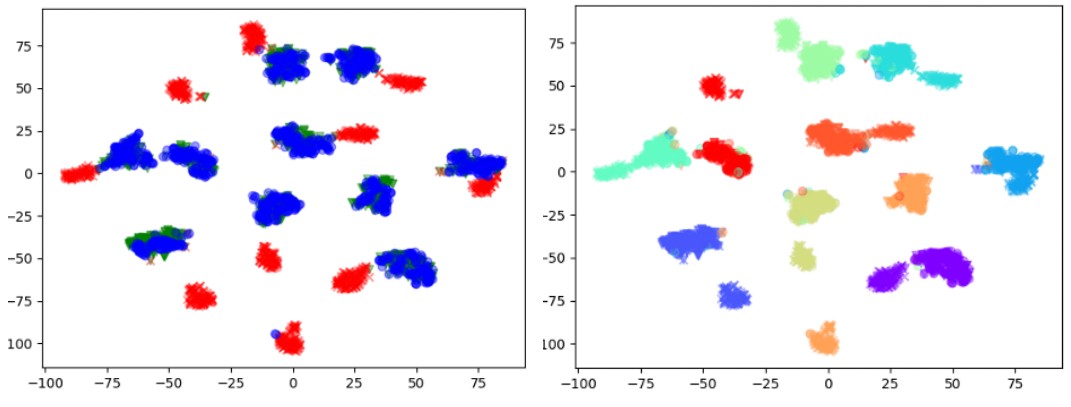

(a) MNIST→USPS balanced

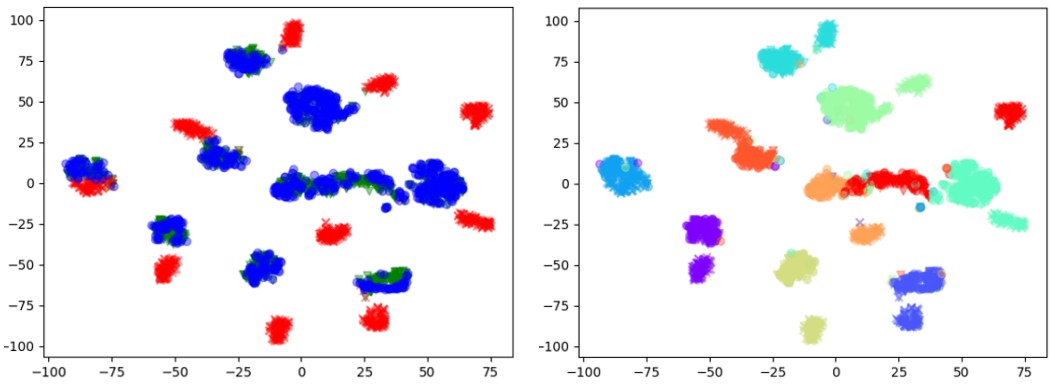

(b) MNIST→USPS subsampled high imbalance

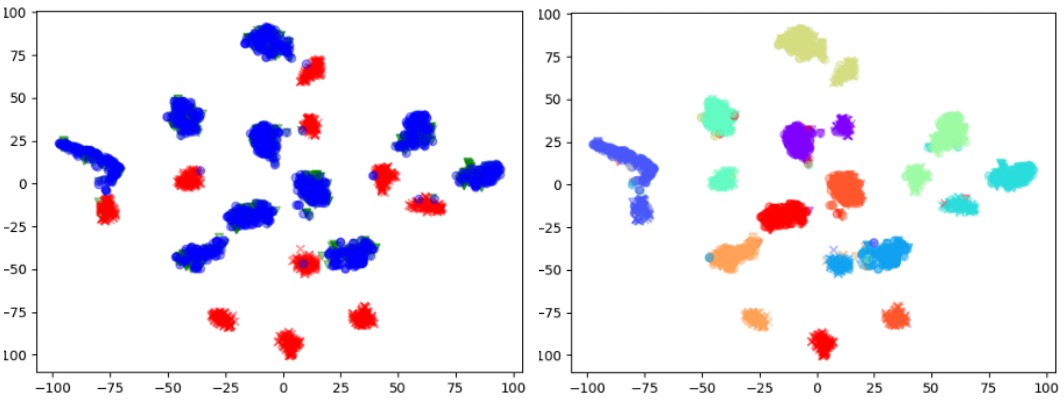

(c) USPS→MNIST balanced

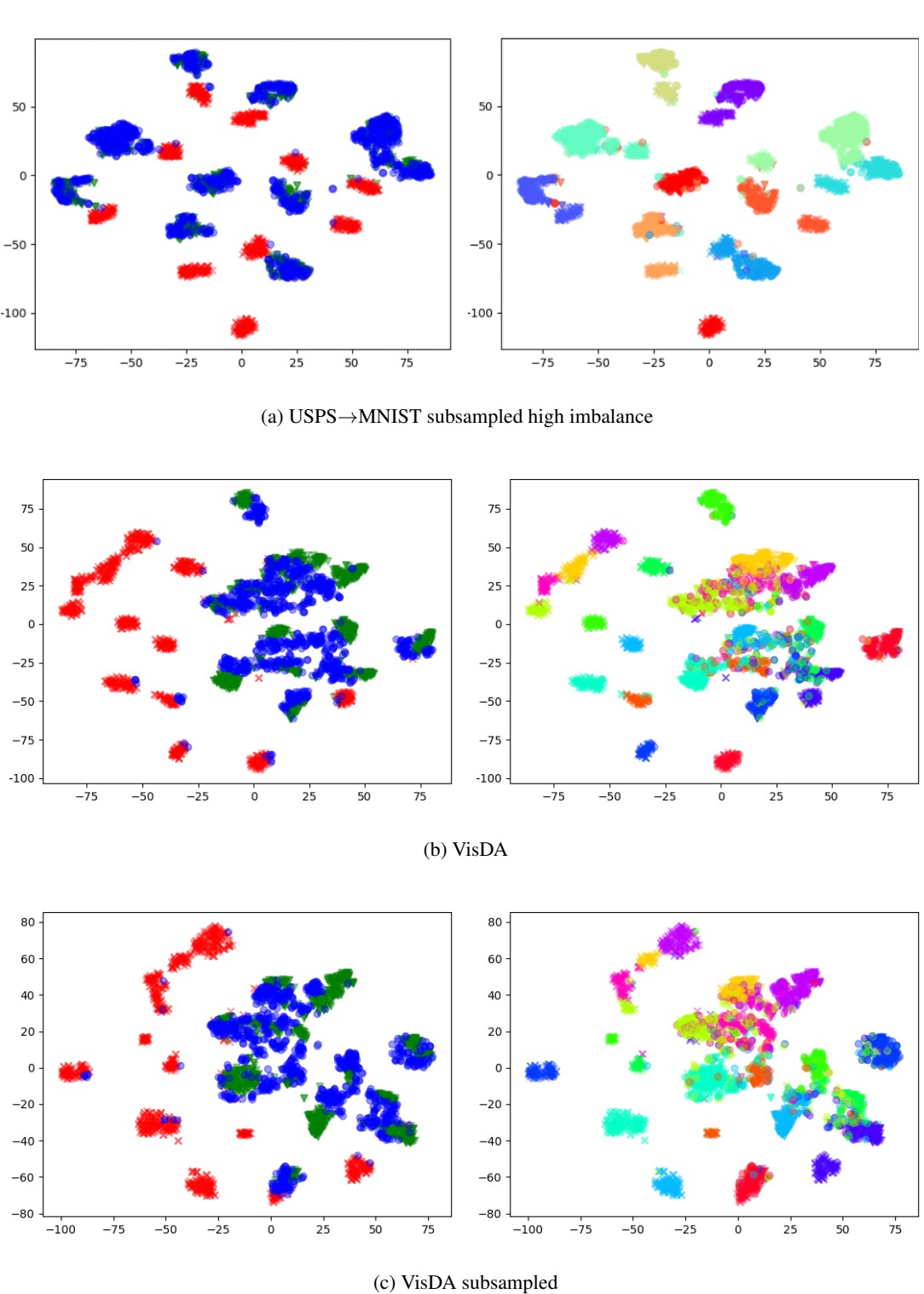

(a) USPS→MNIST subsampled high imbalance

(b) VisDA

(c) VisDA subsampled

Figure 3: t-SNE feature visualizations for OSTAR on various datasets and label imbalance. Crosses "x" denote source samples, circles "o" target samples and triangles "v" transported source samples. On the left, source samples are red, target samples blue and transported source samples green. On the right, samples from the same class have the same colour regardless of domain.

## B  ADDITIONAL RESULTS

We report our full results below. Aggregated results for balanced accuracy over a dataset and imbalance scenarios are reported in Table 1. Prefix "s" refers to subsampled datasets defined in Table 8.

Table 3: Balanced accuracy (↑) over 10 runs. The best performing model is indicated in **bold**.

| Setting | Source | DANN | $WD_{\beta=0}$ | $WD_{\beta=1}$ | $WD_{\beta=2}$ | MARSg | MARSc | IW-WD | OSTAR+IM | DI-Oracle |
|---|---|---|---|---|---|---|---|---|---|---|
| | | | | | Digits - MNIST→USPS | | | | | |
| balanced | $86.94 \pm 1.1$ | $93.97 \pm 0.9$ | $94.42 \pm 0.6$ | $82.05 \pm 6.3$ | $75.14 \pm 6.7$ | $94.19 \pm 1.8$ | $\mathbf{96.44 \pm 0.3}$ | $96.10 \pm 0.3$ | $96.91 \pm 0.3$ | $96.95 \pm 0.2$ |
| mild | $87.10 \pm 2.0$ | $93.23 \pm 1.2$ | $91.52 \pm 0.6$ | $84.07 \pm 5.0$ | $78.29 \pm 7.2$ | $94.78 \pm 1.0$ | $95.18 \pm 0.9$ | $94.72 \pm 0.4$ | $\mathbf{96.18 \pm 1.0}$ | $97.22 \pm 0.2$ |
| high | $86.23 \pm 2.8$ | $93.03 \pm 1.0$ | $91.14 \pm 0.7$ | $85.15 \pm 2.9$ | $78.90 \pm 3.8$ | $94.50 \pm 1.3$ | $95.07 \pm 0.6$ | $94.60 \pm 0.8$ | $\mathbf{96.06 \pm 0.6}$ | $96.87 \pm 0.4$ |
| | | | | | Digits - USPS→MNIST | | | | | |
| balanced | $63.01 \pm 6.5$ | $87.64 \pm 1.7$ | $90.84 \pm 1.3$ | $83.54 \pm 3.0$ | $77.00 \pm 7.6$ | $90.16 \pm 2.5$ | $93.37 \pm 2.5$ | $95.68 \pm 0.6$ | $\mathbf{98.11 \pm 0.2}$ | $96.84 \pm 0.2$ |
| mild | $62.81 \pm 2.3$ | $87.00 \pm 2.1$ | $88.07 \pm 1.2$ | $77.83 \pm 5.4$ | $79.41 \pm 7.4$ | $89.93 \pm 2.4$ | $93.20 \pm 2.8$ | $92.73 \pm 1.5$ | $\mathbf{97.44 \pm 0.5}$ | $95.75 \pm 0.3$ |
| high | $64.04 \pm 5.4$ | $86.37 \pm 1.4$ | $87.04 \pm 1.4$ | $79.17 \pm 6.0$ | $74.47 \pm 7.4$ | $88.24 \pm 3.3$ | $91.54 \pm 0.9$ | $90.81 \pm 1.5$ | $\mathbf{97.08 \pm 0.6}$ | $95.87 \pm 0.3$ |
| | | | | | VisDA12 | | | | | |
| VisDA | $48.63 \pm 1.0$ | $53.72 \pm 0.9$ | $57.40 \pm 1.1$ | $47.56 \pm 0.8$ | $36.21 \pm 1.8$ | $55.62 \pm 1.6$ | $55.33 \pm 0.8$ | $51.88 \pm 1.6$ | $\mathbf{59.24 \pm 0.5}$ | $57.61 \pm 0.3$ |
| sVisDA | $42.46 \pm 1.4$ | $47.57 \pm 0.9$ | $47.32 \pm 1.4$ | $41.48 \pm 1.6$ | $31.83 \pm 3.0$ | $55.00 \pm 1.9$ | $51.86 \pm 2.0$ | $50.65 \pm 1.5$ | $\mathbf{58.84 \pm 1.0}$ | $55.77 \pm 1.1$ |
| | | | | | Office31 | | | | | |
| sA-D | $80.71 \pm 0.5$ | $82.39 \pm 0.4$ | $81.76 \pm 0.4$ | $75.98 \pm 1.2$ | $68.64 \pm 2.4$ | $\mathbf{84.54 \pm 1.0}$ | $84.10 \pm 0.8$ | $81.83 \pm 0.5$ | $84.17 \pm 0.7$ | $87.74 \pm 0.6$ |
| sD-W | $89.08 \pm 0.4$ | $88.70 \pm 0.2$ | $88.98 \pm 0.2$ | $88.53 \pm 0.2$ | $88.97 \pm 0.1$ | $91.03 \pm 0.4$ | $90.76 \pm 0.4$ | $88.17 \pm 0.3$ | $\mathbf{94.13 \pm 0.2}$ | $91.31 \pm 0.2$ |
| sW-A | $58.91 \pm 0.2$ | $58.87 \pm 0.1$ | $59.18 \pm 0.2$ | $60.70 \pm 0.3$ | $60.95 \pm 0.2$ | $63.94 \pm 0.1$ | $63.80 \pm 0.3$ | $60.25 \pm 0.2$ | $\mathbf{69.99 \pm 0.1}$ | $63.92 \pm 0.2$ |
| sW-D | $95.64 \pm 0.2$ | $97.26 \pm 0.3$ | $97.13 \pm 0.3$ | $95.99 \pm 0.3$ | $95.57 \pm 0.5$ | $97.96 \pm 0.1$ | $\mathbf{98.16 \pm 0.2}$ | $97.53 \pm 0.2$ | $98.47 \pm 0.2$ | $98.35 \pm 0.0$ |
| sD-A | $53.41 \pm 0.9$ | $57.45 \pm 0.2$ | $57.81 \pm 0.2$ | $58.24 \pm 0.2$ | $58.61 \pm 0.3$ | $62.12 \pm 0.2$ | $62.13 \pm 0.4$ | $60.03 \pm 0.2$ | $\mathbf{65.00 \pm 0.5}$ | $62.57 \pm 0.3$ |
| sA-W | $69.23 \pm 0.5$ | $72.09 \pm 0.5$ | $72.60 \pm 0.3$ | $65.94 \pm 0.9$ | $61.64 \pm 7.2$ | $81.60 \pm 0.5$ | $81.05 \pm 0.7$ | $75.84 \pm 0.7$ | $\mathbf{83.91 \pm 0.5}$ | $82.51 \pm 0.5$ |
| | | | | | OfficeHome | | | | | |
| sA-C | $44.44 \pm 0.3$ | $46.08 \pm 0.3$ | $41.74 \pm 1.7$ | $40.90 \pm 0.8$ | $39.22 \pm 1.1$ | $47.19 \pm 0.3$ | $46.94 \pm 0.2$ | $45.29 \pm 0.1$ | $\mathbf{48.43 \pm 0.2}$ | $48.09 \pm 0.2$ |
| sA-P | $58.96 \pm 0.3$ | $59.96 \pm 0.2$ | $54.67 \pm 1.8$ | $52.18 \pm 2.3$ | $46.29 \pm 1.4$ | $62.17 \pm 0.2$ | $61.97 \pm 0.2$ | $59.46 \pm 0.3$ | $\mathbf{69.52 \pm 0.4}$ | $63.59 \pm 0.2$ |
| sA-R | $67.10 \pm 0.2$ | $67.42 \pm 0.2$ | $65.40 \pm 0.6$ | $62.52 \pm 1.7$ | $60.51 \pm 1.9$ | $68.66 \pm 0.3$ | $68.62 \pm 0.3$ | $67.76 \pm 0.2$ | $\mathbf{73.29 \pm 0.3}$ | $69.85 \pm 0.1$ |
| sC-A | $35.54 \pm 2.3$ | $35.47 \pm 1.7$ | $37.34 \pm 2.0$ | $36.81 \pm 1.5$ | $33.15 \pm 2.3$ | $46.03 \pm 0.2$ | $\mathbf{46.10 \pm 0.2}$ | $44.18 \pm 0.1$ | $46.47 \pm 0.3$ | $46.94 \pm 0.2$ |
| sC-P | $52.48 \pm 2.1$ | $50.56 \pm 0.9$ | $53.53 \pm 0.1$ | $49.96 \pm 1.4$ | $44.67 \pm 1.5$ | $59.82 \pm 0.1$ | $59.82 \pm 0.1$ | $58.67 \pm 0.1$ | $\mathbf{63.37 \pm 0.1}$ | $60.14 \pm 0.1$ |
| sC-R | $54.99 \pm 1.7$ | $54.22 \pm 0.8$ | $54.69 \pm 0.5$ | $51.34 \pm 2.5$ | $45.16 \pm 3.5$ | $62.69 \pm 0.1$ | $62.41 \pm 0.1$ | $60.74 \pm 0.2$ | $\mathbf{63.12 \pm 0.2}$ | $62.80 \pm 0.2$ |
| sP-A | $38.10 \pm 4.5$ | $36.36 \pm 3.3$ | $48.24 \pm 0.2$ | $47.24 \pm 0.3$ | $48.20 \pm 0.3$ | $47.78 \pm 0.3$ | $46.10 \pm 0.9$ | $45.68 \pm 0.3$ | $\mathbf{50.84 \pm 0.3}$ | $50.11 \pm 0.4$ |
| sP-C | $34.16 \pm 4.6$ | $33.00 \pm 1.9$ | $40.36 \pm 0.4$ | $40.36 \pm 0.4$ | $37.41 \pm 0.3$ | $42.41 \pm 0.3$ | $41.92 \pm 0.3$ | $38.22 \pm 0.2$ | $\mathbf{44.15 \pm 0.3}$ | $43.10 \pm 0.2$ |
| sP-R | $66.28 \pm 3.4$ | $59.19 \pm 0.8$ | $70.01 \pm 0.1$ | $68.78 \pm 0.2$ | $66.61 \pm 0.3$ | $70.00 \pm 0.4$ | $69.37 \pm 0.9$ | $69.43 \pm 0.4$ | $\mathbf{73.95 \pm 0.3}$ | $71.26 \pm 0.4$ |
| sR-P | $66.67 \pm 5.4$ | $70.97 \pm 0.6$ | $73.47 \pm 0.3$ | $72.66 \pm 0.7$ | $71.76 \pm 0.7$ | $72.62 \pm 0.9$ | $72.72 \pm 1.1$ | $72.90 \pm 0.7$ | $\mathbf{75.58 \pm 0.6}$ | $74.17 \pm 0.7$ |
| sR-A | $48.59 \pm 6.7$ | $51.90 \pm 1.1$ | $\mathbf{56.97 \pm 0.3}$ | $57.02 \pm 0.5$ | $55.38 \pm 1.1$ | $54.02 \pm 0.7$ | $53.37 \pm 1.3$ | $53.44 \pm 0.6$ | $56.28 \pm 0.5$ | $57.68 \pm 0.6$ |
| sR-C | $39.36 \pm 2.5$ | $45.33 \pm 0.8$ | $46.47 \pm 0.4$ | $47.11 \pm 0.5$ | $45.38 \pm 1.3$ | $45.81 \pm 1.2$ | $45.30 \pm 1.2$ | $42.66 \pm 1.0$ | $\mathbf{49.07 \pm 0.9}$ | $47.86 \pm 0.3$ |

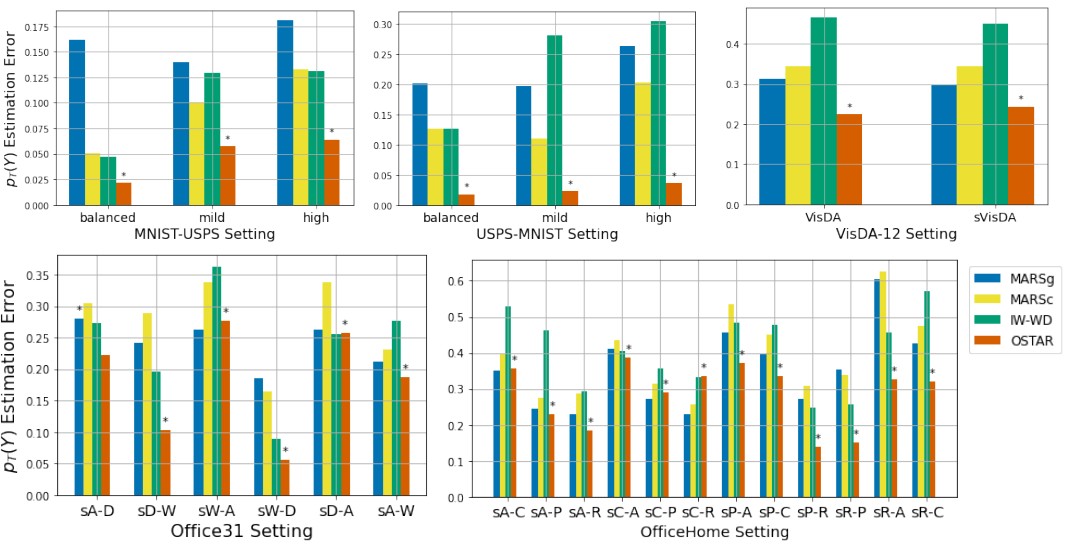

Figure 4: $\ell_1$ estimation error of $\boldsymbol{p}_T^Y$ (↓). The best model for balanced accuracy is indicated with "⋆".

**Additional ablation studies**  We detail the full results for our ablation studies.

Table 4: Semi-supervised learning for OSTAR and balanced accuracy (↑). Best results are in **bold**.

| Setting \ Objective | (CAL) | + (SS) | + (SSg) |
|---|---|---|---|
| MNIST→USPS | | | |
| balanced | $95.12 \pm 0.6$ | $96.68 \pm 0.1$ | $\mathbf{96.91 \pm 0.3}$ |
| mild | $91.77 \pm 1.2$ | $95.39 \pm 1.4$ | $\mathbf{96.18 \pm 1.0}$ |
| high | $88.55 \pm 1.1$ | $95.70 \pm 0.8$ | $\mathbf{96.06 \pm 0.6}$ |
| USPS→MNIST | | | |
| balanced | $88.19 \pm 1.1$ | $97.16 \pm 0.3$ | $\mathbf{98.11 \pm 0.2}$ |
| mild | $88.34 \pm 1.3$ | $96.34 \pm 0.2$ | $\mathbf{97.44 \pm 0.5}$ |
| high | $84.87 \pm 2.3$ | $95.61 \pm 0.4$ | $\mathbf{97.08 \pm 0.6}$ |
| VisDA12 | | | |
| original | $50.37 \pm 0.6$ | $52.54 \pm 0.3$ | $\mathbf{59.24 \pm 0.5}$ |
| subsampled | $49.05 \pm 0.9$ | $53.37 \pm 0.6$ | $\mathbf{58.84 \pm 1.0}$ |
| Office31 | | | |
| sA-D | $81.52 \pm 0.7$ | $83.18 \pm 0.2$ | $\mathbf{84.17 \pm 0.7}$ |
| sD-W | $89.94 \pm 0.8$ | $89.50 \pm 0.8$ | $\mathbf{94.13 \pm 0.2}$ |
| sW-A | $59.62 \pm 0.6$ | $60.06 \pm 0.4$ | $\mathbf{69.99 \pm 0.1}$ |
| sW-D | $96.39 \pm 0.6$ | $97.44 \pm 0.2$ | $\mathbf{98.47 \pm 0.2}$ |
| sD-A | $54.38 \pm 1.1$ | $56.58 \pm 0.6$ | $\mathbf{65.00 \pm 0.5}$ |
| sA-W | $75.30 \pm 1.0$ | $81.32 \pm 0.8$ | $\mathbf{83.91 \pm 0.5}$ |

Table 5: IM for MARSc, IW-WD and OSTAR on balanced accuracy (↑). Best results are in **bold**.

| Setting \ Model | MARSc | MARSc+IM | IW-WD | IW-WD+IM | OSTAR | OSTAR+IM |
|---|---|---|---|---|---|---|
| MNIST→USPS | | | | | | |
| balanced | $96.44 \pm 0.3$ | $\mathbf{97.92 \pm 0.2}$ | $96.10 \pm 0.3$ | $\mathbf{97.91 \pm 0.1}$ | $95.12 \pm 0.6$ | $96.91 \pm 0.3$ |
| mild | $95.18 \pm 0.9$ | $95.47 \pm 1.1$ | $94.72 \pm 0.4$ | $95.74 \pm 0.6$ | $91.77 \pm 1.2$ | $\mathbf{96.18 \pm 1.0}$ |
| high | $95.07 \pm 0.6$ | $93.76 \pm 0.5$ | $94.60 \pm 0.8$ | $91.73 \pm 0.6$ | $88.55 \pm 1.1$ | $\mathbf{96.06 \pm 0.6}$ |
| USPS→MNIST | | | | | | |
| balanced | $93.37 \pm 2.5$ | $93.03 \pm 1.9$ | $95.68 \pm 0.6$ | $96.17 \pm 0.5$ | $88.19 \pm 1.1$ | $\mathbf{98.11 \pm 0.2}$ |
| mild | $93.20 \pm 2.8$ | $94.60 \pm 1.7$ | $92.73 \pm 1.5$ | $92.65 \pm 1.0$ | $88.34 \pm 1.3$ | $\mathbf{97.44 \pm 0.5}$ |
| high | $91.54 \pm 0.9$ | $90.16 \pm 2.0$ | $90.81 \pm 1.5$ | $91.26 \pm 1.1$ | $84.87 \pm 2.3$ | $\mathbf{97.08 \pm 0.6}$ |
| VisDA12 | | | | | | |
| VisDA | $55.33 \pm 0.8$ | $57.57 \pm 0.8$ | $51.88 \pm 1.6$ | $57.63 \pm 0.1$ | $50.37 \pm 0.6$ | $\mathbf{59.24 \pm 0.5}$ |
| sVisDA | $51.86 \pm 2.0$ | $57.06 \pm 0.8$ | $50.65 \pm 1.5$ | $57.62 \pm 0.7$ | $49.05 \pm 0.9$ | $\mathbf{58.84 \pm 1.0}$ |
| Office31 | | | | | | |
| sW-A | $63.80 \pm 0.3$ | $68.12 \pm 0.5$ | $60.25 \pm 0.2$ | $67.42 \pm 0.8$ | $59.62 \pm 0.6$ | $\mathbf{69.99 \pm 0.1}$ |
| sA-W | $81.05 \pm 0.7$ | $81.83 \pm 1.9$ | $75.84 \pm 0.7$ | $82.34 \pm 1.6$ | $75.30 \pm 1.0$ | $\mathbf{83.91 \pm 0.5}$ |
| OfficeHome | | | | | | |
| sR-P | $72.72 \pm 1.1$ | $\mathbf{75.17 \pm 0.6}$ | $72.90 \pm 0.7$ | $74.94 \pm 0.6$ | $71.77 \pm 0.4$ | $\mathbf{75.58 \pm 0.6}$ |
| sR-A | $53.37 \pm 1.3$ | $54.20 \pm 1.3$ | $53.44 \pm 0.6$ | $54.50 \pm 1.1$ | $55.15 \pm 0.8$ | $\mathbf{56.28 \pm 0.5}$ |
| sR-C | $45.30 \pm 1.2$ | $48.17 \pm 1.2$ | $42.66 \pm 1.0$ | $47.93 \pm 1.7$ | $43.02 \pm 3.1$ | $\mathbf{49.07 \pm 0.9}$ |

Table 6: Best value over training epochs of term (A) (↓), term (C) (↓) and term (L) (↓) without and with IM in OSTAR. Best results are in **bold**. Terms (A) and (L) are computed with the primal formulation of OT using the POT package `https://pythonot.github.io/`.

| | Alignment | | | | Discriminativity | |
|---|---|---|---|---|---|---|
| | Term (A) | | Term (L) | | Term (C) | |
| Setting | OSTAR | OSTAR+IM | OSTAR | OSTAR+IM | OSTAR | OSTAR+IM |
| MNIST→USPS | | | | | | |
| balanced | $49.83$ | $\mathbf{16.56}$ | $1.45$ | $\mathbf{0.18}$ | $2.19 \times 10^{-3}$ | $\mathbf{0.918 \times 10^{-3}}$ |
| mild | $39.31$ | $\mathbf{19.13}$ | $10.78$ | $\mathbf{0.24}$ | $1.65 \times 10^{-3}$ | $\mathbf{0.931 \times 10^{-3}}$ |
| high | $38.60$ | $\mathbf{21.10}$ | $12.78$ | $\mathbf{0.74}$ | $1.76 \times 10^{-3}$ | $\mathbf{0.510 \times 10^{-3}}$ |
| USPS→MNIST | | | | | | |
| balanced | $235.43$ | $\mathbf{86.65}$ | $7.08$ | $\mathbf{0.52}$ | $4.46 \times 10^{-3}$ | $\mathbf{0.495 \times 10^{-3}}$ |
| mild | $188.67$ | $\mathbf{104.66}$ | $20.99$ | $\mathbf{1.05}$ | $3.98 \times 10^{-3}$ | $\mathbf{0.399 \times 10^{-3}}$ |
| high | $181.64$ | $\mathbf{123.83}$ | $21.62$ | $\mathbf{0.82}$ | $4.51 \times 10^{-3}$ | $\mathbf{0.616 \times 10^{-3}}$ |

Table 7: Detailed analysis of the impact of $\lambda_{OT}$ on balanced accuracy (↑). Best results are in **bold**.

| | MNIST→USPS - initialization gain 0.02 | | | | | | |
|---|---|---|---|---|---|---|---|
| Shift \ $\lambda_{OT}$ | $\lambda_{OT} = 0$ | $\lambda_{OT} = 10^{-3}$ | $\lambda_{OT} = 10^{-2}$ | $\lambda_{OT} = 10^{-1}$ | $\lambda_{OT} = 1$ | $\lambda_{OT} = 10^4$ | Source |
| balanced | $94.92 \pm 0.6$ | $\mathbf{96.02 \pm 0.2}$ | $95.12 \pm 0.6$ | $89.76 \pm 1.2$ | $91.95 \pm 1.2$ | $87.03 \pm 1.8$ | $86.02 \pm 1.4$ |
| mild | $88.28 \pm 1.5$ | $88.63 \pm 1.3$ | $\mathbf{91.77 \pm 1.2}$ | $90.17 \pm 1.7$ | $88.51 \pm 1.2$ | $88.42 \pm 1.6$ | $89.08 \pm 0.5$ |
| high | $85.24 \pm 1.6$ | $85.38 \pm 1.4$ | $88.55 \pm 1.1$ | $\mathbf{89.10 \pm 1.2}$ | $88.82 \pm 1.1$ | $86.94 \pm 1.1$ | $86.73 \pm 1.9$ |

| Gain \ $\lambda_{OT}$ | $\lambda_{OT} = 0$ | $\lambda_{OT} = 10^{-3}$ | $\lambda_{OT} = 10^{-2}$ | $\lambda_{OT} = 10^{-1}$ | $\lambda_{OT} = 1$ | $\lambda_{OT} = 10^4$ |
|---|---|---|---|---|---|---|
| | | | MNIST→USPS high imbalance | | | |
| 0.02 | $85.24 \pm 1.6$ | $85.38 \pm 1.4$ | $88.55 \pm 1.1$ | $\mathbf{89.10 \pm 1.2}$ | $\mathbf{88.82 \pm 1.1}$ | $86.94 \pm 1.1$ |
| 0.1 | $84.62 \pm 2.3$ | $85.84 \pm 1.1$ | $\mathbf{88.41 \pm 1.3}$ | $\mathbf{88.79 \pm 1.2}$ | $87.90 \pm 1.2$ | $87.10 \pm 1.0$ |
| 0.3 | $83.11 \pm 2.4$ | $84.45 \pm 1.4$ | $89.41 \pm 1.6$ | $\mathbf{91.00 \pm 1.3}$ | $89.65 \pm 0.7$ | $86.23 \pm 1.8$ |

## C  DETAILS ON OPTIMAL TRANSPORT

**Background**   OT was introduced to find a transportation map minimizing the cost of displacing mass from one configuration to another (Villani, 2008). For a comprehensive introduction, we refer to Peyré & Cuturi (2019). Formally, let $\alpha$ and $\beta$ be absolutely continuous distributions compactly supported in $\mathbb{R}^d$ and $c : \mathbb{R}^d \times \mathbb{R}^d \to \mathbb{R}$ a cost function. Consider a map $\phi : \mathbb{R}^d \to \mathbb{R}^d$ that satisfies $\phi_\# \alpha = \beta$, i.e. that pushes $\alpha$ to $\beta$. We remind that for a function $f$, $f_\# \rho$ is the push-forward measure $f_\# \rho(B) = \rho\left(f^{-1}(B)\right)$, for all measurable set $B$. The total transportation cost depends on the contributions of costs for transporting each point $\mathbf{x}$ to $\phi(\mathbf{x})$ and the Monge OT problem is:

$$\min_\phi \mathcal{C}_{\text{monge}}(\phi) = \int_{\mathbb{R}^d} c(\mathbf{x}, \phi(\mathbf{x})) d\alpha(\mathbf{x}) \tag{9}$$
$$\text{s.t. } \phi_\# \alpha = \beta$$

$c(\mathbf{x}, \mathbf{y}) = \|\mathbf{x} - \mathbf{y}\|_2^p$ induces the $p$-Wasserstein distance, $\mathcal{W}_p(\alpha, \beta) = \min_{\phi_\# \alpha = \beta} \mathcal{C}_{\text{monge}}(\phi)^{1/p}$. When $p = 1$, $\mathcal{W}_1$ can be expressed in the dual form $\mathcal{W}_1(\alpha, \beta) = \sup_{\|v\|_L \leq 1} \mathbb{E}_{\mathbf{x} \sim \alpha} v(\mathbf{x}) - \mathbb{E}_{\mathbf{y} \sim \beta} v(\mathbf{y})$ where $\|v\|_L$ is the Lipschitz constant of function $v$.

**Relationship between** (OT) **and the Monge OT problem** (9)    (OT) is the extension of (9) to the setting where $\boldsymbol{p}_S^Y \neq \boldsymbol{p}_T^Y$ with $\boldsymbol{p}_T^Y$ unknown. This extension aims at matching conditional distributions regardless of how their mass differ and learns a weighting term $\boldsymbol{p}_N^Y$ for source conditional distributions which addresses label shift settings. When $\boldsymbol{p}_S^Y = \boldsymbol{p}_T^Y$, these two formulations are equivalent under Assumption 1 if we fix $\boldsymbol{p}_N^Y = \boldsymbol{p}_S^Y = \boldsymbol{p}_T^Y$.

## D  DISCUSSION

Our four assumptions are required for deriving our theoretical guarantees. All `GeTarS` papers that propose theoretical guarantees also rely on a set of assumptions, either explicit or implicit. Assumptions 1 and 4 are common to several papers. Assumption 1 is easily met in practice since it could be forced by training a classifier on the source labels. Assumption 4 is said to be met in high dimensions (Redko et al., 2019; Garg et al., 2020). Assumption 3 says that source and target clusters from the same class are globally (there is a sum in the condition) closer one another than clusters from two different classes. This assumption is required to cope with the absence of target labels. Because of the sum, it could be considered as a reasonable hypothesis. It is milder than the hypotheses made in papers offering guarantees similar to ours (Zhang et al., 2013; Gong et al., 2016). Assumption 2 is new to this paper. It states that $\phi$ maps a $j$ source conditional to a unique $k$ target conditional i.e. it guarantees that mass of a source conditional will be entirely transferred to a target conditional and will not be split across several target conditionals. This property is key to show that `OSTAR` matches source conditionals and their corresponding target conditionals, which is otherwise very difficult to guarantee under `GeTarS` as both target conditionals and proportions are unknown. This assumption has some restrictions, despite being milder than existing assumptions in `GeTarS`. First, mass from a source conditional might in practise be split across several target conditionals. In practise, our optimization problem in (CAL) mitigates this problem by learning how to reweight source conditionals to improve alignment. We could additionally apply Laplacian regularization (Ferradans et al., 2013; Carreira-Perpiñán & Wang, 2014) into our OT map $\phi$ but this was not developed in the paper. Second, it assumes a closed-set UDA problem i.e. label domains are the same $\mathcal{Y}_S = \mathcal{Y}_T$. The setting where $\mathcal{Y}_S \neq \mathcal{Y}_T$ is more realistic in large scale image pretraining and is another interesting follow-up. Note that `OSTAR` can address empirically open-set UDA ($\mathcal{Y}_T \subset \mathcal{Y}_S$) by simply applying L1 regularization on $\boldsymbol{p}_N^Y$ in our objective function (CAL). This forces sparsity and allows "loosing" mass when mapping a source conditional. It avoids the unwanted negative transfer setting when source clusters, with labels absent on the target, are aligned with target clusters.

## E  PROOFS

**Proposition (1).** *For any encoder g which defines $\mathcal{Z}$ satisfying Assumption 1, 2, 3, 4, there is an unique solution $(\phi, \boldsymbol{p}_N^Y)$ to (OT) and $\phi_{\#}(p_S(Z|Y)) = p_T(Z|Y)$ and $\boldsymbol{p}_N^Y = \boldsymbol{p}_T^Y$.*

*Proof.* Fixing $\mathcal{Z}$ satisfying Assumption 1, 3 and 4, we first show that there exists a solution $(\phi, \boldsymbol{p}_N^Y)$ to (OT). Following Brenier (1991) as $\mathcal{Z} \subset \mathbb{R}^d$, we can find $K$ unique Monge maps $\{\widehat{\phi}^{(k)}\}_{k=1}^K$ s.t. $\forall k \ \widehat{\phi}_{\#}^{(k)}(p_S(Z|Y = k)) = p_T(Z|Y = k)$ with respective transport costs $\mathcal{W}_1(p_S(Z|Y = k), p_T(Z|Y = k))$. Let's define $\widehat{\phi}$ as $\forall k \ \widehat{\phi}_{|\mathcal{Z}_S^{(k)}} = \widehat{\phi}^{(k)}$ where $\cup_{k=1}^K \mathcal{Z}_S^{(k)}$ is the partition of $\mathcal{Z}_S$ in Assumption 1. $(\widehat{\phi}, \boldsymbol{p}_T^Y)$ satisfies the equality constraint to (OT), thus we easily deduce existence.

Now, let $(\phi, \boldsymbol{p}_N^Y)$ be a solution to (OT), let's show unicity. We first show that $\phi = \widehat{\phi}$. Under Assumption 2, (OT) is the Monge formulation of the optimal assignment problem between $\{p_S(Z|Y = k)\}_{k=1}^K$ and $\{p_T(Z|Y = k)\}_{k=1}^K$ with $\mathbf{C}$ the cost matrix defined by $\mathbf{C}_{ij} = \mathcal{W}_2(p_S(Z|Y = i), p_T(Z|Y = j))$. At the optimum, the transport cost is related to the Wasserstein distance between source conditionals and their corresponding target conditionals i.e. $\mathcal{C}(\phi) = \sum_{k=1}^K \mathcal{W}_2(p_S(Z|Y = k), \phi_{\#}(p_S(Z|Y = k)))$. Suppose $\exists (i,j), j \neq i$ s.t. $\phi_{\#}(p_S(Z|Y = i)) = p_T(Z|Y = j)$ and $\phi_{\#}(p_S(Z|Y = j)) = p_T(Z|Y = i)$ and $\forall k \neq i, j, \phi_{\#}(p_S(Z|Y = k)) = p_T(Z|Y = k)$. Assumption 3 implies $\sum_{k=1}^K \mathcal{W}_2(p_S(Z|Y = k), p_T(Z|Y = k)) \leq \sum_{k \neq i,j} \mathcal{W}_2(p_S(Z|Y = k), p_T(Z|Y = k)) + \mathcal{W}_2(p_S(Z|Y = i), p_T(Z|Y = j)) + \mathcal{W}_2(p_S(Z|Y = j), p_T(Z|Y = i))$. Thus $C(\widehat{\phi}, \mathcal{Z}) \leq C(\phi, \mathcal{Z})$ whereas $\phi$, solution to (OT), has minimal transport cost. Thus $\phi = \widehat{\phi}$.

Now let's show $\boldsymbol{p}_N^Y = \boldsymbol{p}_T^Y$ under Assumption 4. We inject $\phi_{\#}(p_S(Z|Y)) = p_T(Z|Y)$ into (OT),

$$\sum_{k=1}^K \boldsymbol{p}_N^{Y=k} p_T(Z|k) = \sum_{k=1}^K \boldsymbol{p}_T^{Y=k} p_T(Z|k) \Leftrightarrow \sum_{k=1}^K \left(\boldsymbol{p}_N^{Y=k} - \boldsymbol{p}_T^{Y=k}\right) p_T(Z|k) = 0 \Leftrightarrow \boldsymbol{p}_N^Y = \boldsymbol{p}_T^Y$$

$\square$

**Theorem (1).** *Given a fixed encoder g defining a latent space $\mathcal{Z}$, two domains $N$ and $T$ satisfying cyclical monotonicity in $\mathcal{Z}$, assuming that we have $\forall k, \boldsymbol{p}_N^{Y=k} > 0$, then $\forall f_N \in \mathcal{H}$ where $\mathcal{H}$ is a set of $M$-Lipschitz continuous functions over $\mathcal{Z}$, we have*

$$\epsilon_T^g(f_N) \leq \underbrace{\epsilon_N^g(f_N)}_{\text{Classification } (C)} + \underbrace{\frac{2M}{\min_{k=1}^K \boldsymbol{p}_N^{Y=k}} \mathcal{W}_1\Big(p_N(Z), p_T(Z)\Big)}_{\text{Alignment } (A)}$$

$$+ \underbrace{2M\Big(1 + \frac{1}{\min_{k=1}^K \boldsymbol{p}_N^{Y=k}}\Big) \mathcal{W}_1\Big(\sum_{k=1}^K \boldsymbol{p}_N^{Y=k} p_T(Z|Y = k), \sum_{k=1}^K \boldsymbol{p}_T^{Y=k} p_T(Z|Y = k)\Big)}_{\text{Label } (L)} \quad (2)$$

*Proof.* We first recall that $\epsilon_D^g(f_N) = \mathbb{E}_{(\mathbf{z},y) \in p_D(Z,Y)} \mathcal{L}(f_N(\mathbf{z}), y)$ where $\mathcal{L}$ is the 0/1 loss. For conciseness, $\forall \mathbf{z} \in \mathcal{Z}, \boldsymbol{p}_T^{\mathbf{z}|Y}, \boldsymbol{p}_N^{\mathbf{z}|Y}, \mathcal{L}^{\mathbf{z},k}$ will refer to the vector of $[p_T(\mathbf{z}|k)]_{k=1}^K$, $[p_N(\mathbf{z}|k)]_{k=1}^K$, $[\mathcal{L}(f_N(\mathbf{z}), k)]_{k=1}^K$ respectively. In the following, $\odot$ denotes the element-wise product operator between two vectors of the same size.

$$\forall f_N, \epsilon_T^g(f_N) = \epsilon_N^g(f_N) + \epsilon_T^g(f_N) - \epsilon_N^g(f_N)$$

$$\leq \epsilon_N^g(f_N) + \int_{\mathcal{Z}} \sum_{k=1}^K \Big(p_T(\mathbf{z},k) - p_N(\mathbf{z},k)\Big) \times \mathcal{L}\Big(f_N(\mathbf{z}), k\Big) d\mathbf{z} dy$$

$$\leq \epsilon_N^g(f_N) + \int_{\mathcal{Z}} \sum_{k=1}^K \Big[\boldsymbol{p}_T^{Y=k} p_T(\mathbf{z}|k) - \boldsymbol{p}_N^{Y=k} p_N(\mathbf{z}|k)\Big] \times \mathcal{L}\Big(f_N(\mathbf{z}), k\Big) d\mathbf{z}$$

$$\leq \epsilon_N^g(f_N) + \int_{\mathcal{Z}} \boldsymbol{p}_T^{Y\mathsf{T}}\Big(\boldsymbol{p}_T^{\mathbf{z}|Y} \odot \mathcal{L}^{\mathbf{z},k}\Big) - \boldsymbol{p}_N^{Y\mathsf{T}}\Big(\boldsymbol{p}_N^{\mathbf{z}|Y} \odot \mathcal{L}^{\mathbf{z},k}\Big) d\mathbf{z}$$

$$\leq \epsilon_N^g(f_N) + \int_{\mathcal{Z}} \boldsymbol{p}_T^{Y\mathsf{T}}\Big(\boldsymbol{p}_T^{\mathbf{z}|Y} \odot \mathcal{L}^{\mathbf{z},k}\Big) - \boldsymbol{p}_N^{Y\mathsf{T}}\Big(\boldsymbol{p}_T^{\mathbf{z}|Y} \odot \mathcal{L}^{\mathbf{z},k}\Big) + \boldsymbol{p}_N^{Y\mathsf{T}}\Big(\boldsymbol{p}_T^{\mathbf{z}|Y} \odot \mathcal{L}^{\mathbf{z},k}\Big) - \boldsymbol{p}_N^{Y\mathsf{T}}\Big(\boldsymbol{p}_{\boldsymbol{N}}^{\mathbf{z}|Y} \odot \mathcal{L}^{\mathbf{z},k}\Big) d\mathbf{z}$$

$$\leq \epsilon_N^g(f_N) + \int_{\mathcal{Z}} \Big(\boldsymbol{p}_T^{Y\mathsf{T}} - \boldsymbol{p}_N^{Y\mathsf{T}}\Big)\Big(\boldsymbol{p}_T^{\mathbf{z}|Y} \odot \mathcal{L}^{\mathbf{z},k}\Big) + \boldsymbol{p}_N^{Y\mathsf{T}}\Big((\boldsymbol{p}_T^{\mathbf{z}|Y} - \boldsymbol{p}_{\boldsymbol{N}}^{\mathbf{z}|Y}) \odot \mathcal{L}^{\mathbf{z},k}\Big) d\mathbf{z}$$

$$\leq \epsilon_N^g(f_N) + \int_{\mathcal{Z}} \Big(\boldsymbol{p}_T^{Y\mathsf{T}} - \boldsymbol{p}_N^{Y\mathsf{T}}\Big)\Big(\boldsymbol{p}_T^{\mathbf{z}|Y} \odot \mathcal{L}^{\mathbf{z},k}\Big) d\mathbf{z} + \int_{\mathcal{Z}} \boldsymbol{p}_N^{Y\mathsf{T}}\Big((\boldsymbol{p}_T^{\mathbf{z}|Y} - \boldsymbol{p}_{\boldsymbol{N}}^{\mathbf{z}|Y}) \odot \mathcal{L}^{\mathbf{z},k}\Big) d\mathbf{z}$$

We now introduce a preliminary result from Shen et al. (2018). $\forall f_N \in \mathcal{H}$ $M$-Lipschitz continuous,

$$\epsilon_N^g(f_N) - \epsilon_T^g(f_N) \leq 2M \cdot \mathcal{W}_1(p_N^g(Z), p_T^g(Z))$$

Assuming that $h$ is $M$-Lipschitz continuous we apply this result in the following

$$\int_{\mathcal{Z}} \Big(\boldsymbol{p}_T^{Y\mathsf{T}} - \boldsymbol{p}_N^{Y\mathsf{T}}\Big)\Big(\boldsymbol{p}_T^{\mathbf{z}|Y} \odot \mathcal{L}^{\mathbf{z},k}\Big) d\mathbf{z} = \int_{\mathcal{Z}} \sum_{k=1}^{K} \Big(\boldsymbol{p}_T^{Y=k} - \boldsymbol{p}_N^{Y=k}\Big) p_T(\mathbf{z}|k) \times \mathcal{L}\Big(f_N(\mathbf{z}), k\Big) d\mathbf{z}$$

$$= \epsilon_T^g(f_N) - \epsilon_{\widetilde{T}}^g(f_N) \quad \text{where } p_{\widetilde{T}}(Z) = \sum_{k=1}^{K} \boldsymbol{p}_N^{Y=k} p_T(Z|k)$$

$$\leq 2M \cdot \mathcal{W}_1(p_{\widetilde{T}}(Z), p_T(Z))$$

$$\int_{\mathcal{Z}} \boldsymbol{p}_N^{Y\mathsf{T}}\Big((\boldsymbol{p}_T^{\mathbf{z}|Y} - \boldsymbol{p}_{\boldsymbol{N}}^{\mathbf{z}|Y}) \odot \mathcal{L}^{\mathbf{z},k}\Big) d\mathbf{z} = \int_{\mathcal{Z}} \sum_{k=1}^{K} \boldsymbol{p}_N^{Y=k}\Big(p_T(\mathbf{z}|k) - p_N(\mathbf{z}|k)\Big) \times \mathcal{L}\Big(f_N(\mathbf{z}), k\Big) d\mathbf{z}$$

$$\leq \sum_{k=1}^{K} \int_{\mathcal{Z}} \Big(p_T(\mathbf{z}|k) - p_N(\mathbf{z}|k)\Big) \times \mathcal{L}\Big(f_N(\mathbf{z}), k\Big) d\mathbf{z} \quad \forall k \; \boldsymbol{p}_N^{Y=k} \leq 1$$

$$\leq 2M \sum_{k=1}^{K} \mathcal{W}_1\Big(p_T(Z|k), p_N(Z|k)\Big)$$

Thus, $\forall f_N$ $M$-Lipschitz continuous

$$\epsilon_T^g(f_N) \leq \epsilon_N^g(f_N) + 2M \times \sum_{k=1}^{K} \mathcal{W}_1\Big(p_T(Z|k), p_N(Z|k)\Big) + 2M \times \mathcal{W}_1\Big(p_{\widetilde{T}}(Z), p_T(Z)\Big)$$

We rewrite the second term to involve directly latent marginals. Proposition 2 in Rakotomamonjy et al. (2021) shows that under cyclical monotonicity, if $\forall k, \boldsymbol{p}_N^{Y=k} > 0$,

$$\mathcal{W}_1\Big(\sum_{k=1}^{K} \boldsymbol{p}_N^{Y=k} p_T(Z|k), p_N(Z)\Big) = \sum_{k=1}^{K} \boldsymbol{p}_N^{Y=k} \mathcal{W}_1\Big(p_N(Z|k), p_T(Z|k)\Big)$$

This allows to write

$$\min_{k=1}^{K} \boldsymbol{p}_N^{Y=k} \sum_{k=1}^{K} \mathcal{W}_1\Big(p_N(Z|k), p_T(Z|k)\Big) \leq \sum_{k=1}^{K} \boldsymbol{p}_N^{Y=k} \mathcal{W}_1\Big(p_N(Z|k), p_T(Z|k)\Big)$$

$$= \mathcal{W}_1\Big(\sum_{k=1}^{K} \boldsymbol{p}_N^{Y=k} p_T(Z|k), p_N(Z)\Big) = \mathcal{W}_1\Big(p_{\widetilde{T}}(Z), p_N(Z)\Big)$$

We then use the triangle inequality for the Wasserstein distance $\mathcal{W}_1$

$$\epsilon_T^g(f_N) \leq \epsilon_N^g(f_N) + \frac{2M}{\min_{k=1}^{K} \boldsymbol{p}_N^{Y=k}} \mathcal{W}_1\Big(p_{\widetilde{T}}(Z), p_N(Z)\Big) + 2M \times \mathcal{W}_1\Big(p_{\widetilde{T}}(Z), p_T(Z)\Big)$$

$$\leq \epsilon_N^g(f_N) + \frac{2M}{\min_{k=1}^{K} \boldsymbol{p}_N^{Y=k}} \mathcal{W}_1\Big(p_N(Z), p_T(Z)\Big) + 2M(1 + \frac{1}{\min_{k=1}^{K} \boldsymbol{p}_N^{Y=k}}) \mathcal{W}_1\Big(p_{\widetilde{T}}(Z), p_T(Z)\Big)$$

$$\square$$

**Derivation of the reweighted classification loss (C)** $\epsilon_N^g(f_N)$    Let $\mathcal{L}_{ce}$ be the cross-entropy loss. Given a classifier $h$, feature extractor $g$ and domain $N$, the mapping of domain $S$ by $(\phi, \boldsymbol{p}_N^Y)$,

$$\epsilon_N^g(f_N) = \int_{\mathcal{Z},\mathcal{Y}} p_N^\phi(\mathbf{z}, y)\mathcal{L}_{ce}\Big(f_N(\mathbf{z}), y\Big)d\mathbf{z}dy = \int_{\mathcal{Z},\mathcal{Y}} \boldsymbol{p}_N^{Y=y}p_N^\phi(\mathbf{z}|y)\mathcal{L}_{ce}\Big(f_N(\mathbf{z}), y\Big)d\mathbf{z}dy$$

$$= \int_{\mathcal{Z},\mathcal{Y}} \frac{\boldsymbol{p}_N^{Y=y}}{\boldsymbol{p}_S^{Y=y}}\boldsymbol{p}_S^{Y=y}p_N^\phi(\mathbf{z}|y)\mathcal{L}_{ce}\Big(f_N(\mathbf{z}), y\Big)d\mathbf{z}dy = \int_{\mathcal{Z},\mathcal{Y}} \frac{\boldsymbol{p}_N^{Y=y}}{\boldsymbol{p}_S^{Y=y}}\boldsymbol{p}_S^{Y=y}\phi_\#(p_S(\mathbf{z}|y))\mathcal{L}_{ce}\Big(f_N(\mathbf{z}), y\Big)d\mathbf{z}dy$$

## F   PSEUDO-CODE AND RUNTIME / COMPLEXITY ANALYSIS

We detail in Algorithm 1 our pseudo-code and in Algorithm 2 how we minimize (CAL) with respect to $(\phi, f_N)$ using the dual form of Wasserstein-1 distance (6). Our method is based on a standard backpropagation strategy with gradient descent and uses gradient penalty (Gulrajani et al., 2017).

---

**Algorithm 1** Training and inference procedure for OSTAR

---

**Training**:
$\widehat{S} = \{\mathbf{x}_\mathbf{S}^{(i)}, y_S^{(i)}\}_{i=1}^n, \widehat{T} = \{\mathbf{x}_\mathbf{T}^{(i)}\}_{i=1}^m, \mathcal{Z}_{\widehat{N}} = \{\phi \circ g(\mathbf{x}_\mathbf{S}^{(i)}), y_S^{(i)}\}_{i=1}^n, \mathcal{Z}_{\widehat{T}} = \{g(\mathbf{x}_\mathbf{T}^{(i)})\}_{i=1}^m$
$f_S, f_N \in \mathcal{H}$ classifiers; $g$ feature extractor; $\phi$ latent domain-mapping, $v$ critic.
$N_e$: number of epochs, $N_u$: epoch to update $\boldsymbol{p}_N^Y$, $N_g$: epoch to update $g$

1: Train $f_S, g$ on $\widehat{S}$ to minimize source classification loss                                    ▷ (3)
2: Initialize $\boldsymbol{p}_N^Y = \frac{1}{K}\mathbf{1}_K$
3: **for** $n_{epoch} \leq N_e$ **do**
4:     **if** $n_{epoch} \mod N_u = 0$ **then**
5:         Compute $\boldsymbol{p}_N^Y$ with estimator in Lipton et al. (2018) on $(\mathcal{Z}_{\widehat{N}}, \mathcal{Z}_{\widehat{T}})$        ▷ (CAL) w.r.t. $\boldsymbol{p}_N^Y$
6:         Average $\boldsymbol{p}_N^Y$ with cumulative moving average
7:     **if** $n_{epoch} \leq N_g$ **then** Train $\phi, v, f_N$ with $(\widehat{S}, \widehat{T})$        ▷ (CAL) + (SS) w.r.t. $\phi, f_N$
8:     **else** Train $\phi, v, f_N, g$ with $(\widehat{S}, \widehat{T})$        ▷ (CAL) + (SSg) w.r.t. $\phi, f_N$

**Inference:** Score $\mathbf{x}_\mathbf{T}$ with $f_N \circ g(\mathbf{x}_\mathbf{T})$

---

**Algorithm 2** Minimize (CAL) w.r.t. $(\phi, f_N)$

---

$\widehat{S} = \{\mathbf{x}_\mathbf{S}^{(i)}, y_S^{(i)}\}_{i=1}^n, \widehat{T} = \{\mathbf{x}_\mathbf{T}^{(i)}\}_{i=1}^m, g$ feature extractor, $\phi$ domain-mapping, $v$ critic, $f_N$ classifier.
Parameters of $\phi, v, f_N$: $\theta_\phi, \theta_v, \theta_{f_N}$ and learning rates $\alpha_\phi, \alpha_v, \alpha_{f_N}$.
$N_{iter}$: batches per epoch, $N_b$: batch size, $N_v$: critic iterations

1: **for** $n_{iter} < N_{iter}$ **do**
2:     Sample minibatches $\mathbf{x}_\mathbf{S}^\mathbf{B}, y_S^B = \{\mathbf{x}_\mathbf{S}^{(i)}, y_S^{(i)}\}_{i=1}^{N_b}, \mathbf{x}_\mathbf{T}^\mathbf{B} = \{\mathbf{z}_\mathbf{T}^{(i)}\}_{i=1}^{N_b}$ from $\widehat{S}, \widehat{T}$
3:     Compute $\mathbf{z}_\mathbf{S}^\mathbf{B} = g(\mathbf{x}_\mathbf{S}^\mathbf{B}), \mathbf{z}_\mathbf{N}^\mathbf{B} = \phi \circ g(\mathbf{x}_\mathbf{S}^\mathbf{B})$ and $\mathbf{z}_\mathbf{T}^\mathbf{B} = g(\mathbf{x}_\mathbf{T}^\mathbf{B})$
4:     Compute class ratios: $\mathbf{w}_Y = \boldsymbol{p}_N^Y/\boldsymbol{p}_S^Y$
5:     **for** $n_v < N_v$ **do**
6:         Sample random points $\mathbf{z}^{\mathbf{B}'}$ from the lines between $(\mathbf{z}_\mathbf{N}^\mathbf{B}, \mathbf{z}_\mathbf{T}^\mathbf{B})$ pairs
7:         Compute gradient penalty $\mathcal{L}_{grad}$ with $\mathbf{z}_\mathbf{N}^\mathbf{B}, \mathbf{z}_\mathbf{T}^\mathbf{B}, \mathbf{z}^{\mathbf{B}'}$ (Gulrajani et al., 2017)
8:         Compute $\mathcal{L}_{wd}^g = \sum_{i=1}^{N_b} \mathbf{w}_{y_S^{(i)}} v(\mathbf{z}_\mathbf{N}^{(i)}) - \frac{1}{N_b}\sum_{i=1}^{N_b} v(\mathbf{z}_\mathbf{T}^{(i)})$ (6)
9:         $\theta_v \leftarrow \theta_v - \alpha_v \nabla_{\theta_v}\Big[\mathcal{L}_{wd}^g - \mathcal{L}_{grad}\Big]$
10:     Compute $\mathcal{L}_{OT}^g = \sum_{k=1}^K \frac{1}{\#\{y_S^{(i)} = k\}_{i \in [\![1,N_b]\!]}} \sum_{y_S^{(i)}=k, \ i \in [\![1,N_b]\!]} \|\phi(\mathbf{z}_\mathbf{S}^{(i)}) - \mathbf{z}_\mathbf{S}^{(i)}\|_2^2$
11:     $\theta_\phi \leftarrow \theta_\phi - \alpha_\phi \nabla_{\theta_\phi}\Big[\mathcal{L}_{wd}^g + \mathcal{L}_{OT}^g\Big]$
12:     Compute $\mathcal{L}_c^g(f_N, N) = \frac{1}{N_b}\sum_{i=1}^{N_b} \mathbf{w}_{y_S^{(i)}}\mathcal{L}_{ce}(f_N \circ \phi \circ g(\mathbf{x}_\mathbf{S}^{(i)}), y_S^{(i)})$
13:     $\theta_{f_N} \leftarrow \theta_{f_N} - \alpha_{f_N} \nabla_{\theta_{f_N}}\mathcal{L}_c^g(f_N, N)$

---

**Complexity / Runtime analysis**   In practise on USPS → MNIST, the runtimes in seconds on a NVIDIA Tesla V100 GPU machine are the following: DANN: 22.75s, WD$_{\beta=0}$: 59.25s, MARSc: 72.87s, MARSg: 2769.06s, IW-WD: 74.17s, OSTAR+IM: 89.72s. We observe that (i) computing Wasserstein distance (WD$_{\beta=0}$) is slower than computing $\mathcal{H}$-divergence (DANN), (ii) runtimes for domain-invariant GeTarS baselines are slightly higher than for WD$_{\beta=0}$ as proportions are additionally estimated, (iii) domain-invariant GeTarS baselines have similar runtimes apart from MARSg which uses GMM, (iv) our model, OSTAR+IM, has slightly higher runtime than GeTarS baselines other than MARSg. We now provide some further analysis on computational cost and memory in OSTAR. We denote $K$ the number of classes, $d$ the dimension of latent representations, $n$ and $m$ the number of source and target samples.

- Memory: Proportion estimation is based on the method in Lipton et al. (2018) and requires storing the confusion matrix $\hat{\mathbf{C}}$ and predictions $\hat{\boldsymbol{p}}_T^Y$ with a memory cost of $O(K^2)$. Encoding is performed by a deep NN $g$ into $\mathbb{R}^d$ and classification by a shallow classifier from $\mathbb{R}^d$ to $\mathbb{R}^K$. Alignment is performed with a ResNet $\phi : \mathbb{R}^d \to \mathbb{R}^d$ in the latent space which has an order magnitude less parameters than $g$. Globally, memory consumption is mostly defined by the number of parameters in the encoder $g$.

- Computational cost comes from: (i) proportion estimation, which depends on $K, n+m$ and is solved once every 5 epochs with small runtime; (ii) alignment between source and target representations, which depends on $d, n + m$ and $N_v$ (the number of critic iterations). This step updates less parameters than domain-invariant methods which align with $g$ instead of $\phi$; this may lead to speedups for large encoders. The transport cost $\mathcal{L}_{OT}^g$ depends on $n, d$ and adds small additional runtime; (iii) classification with $f_N$ using labelled source samples, which depends on $d, K, n$. In a second stage, we improve target discriminativity by updating the encoder $g$ with semi-supervised learning; this depends on $d, K, n + m$.

# G   EXPERIMENTAL SETUP

**Datasets**   We consider the following UDA problems:

- Digits is a synthetic binary adaptation problem. We consider adaptation between MNIST and USPS datasets. We consider a subsampled version of the original datasets with the following number of samples per domain: 10000-2609 for MNIST→USPS, 5700-20000 for USPS→MNIST. The feature extractor is learned from scratch.

- VisDA12 is a 12-class adaptation problem between simulated and real images. We consider a subsampled version of the original problem using 9600 samples per domain and use pre-trained ImageNet ResNet-50 features http://csr.bu.edu/ftp/visda17/clf/.

- Office31 is an object categorization problem with 31 classes. We do not sample the original dataset. There are 3 domains: Amazon (A), DSLR (D) and WebCam (W) and we consider all pairwise source-target domains. We use pre-trained ImageNet ResNet-50 features https://github.com/jindongwang/transferlearning/blob/master/data/dataset.md.

- OfficeHome is another object categorization problem with 65 classes. We do not sample the original dataset. There are 4 domains: Art (A), Product (P), Clipart (C), Realworld (R) and we consider all pairwise source-target domains. We use pre-trained ImageNet ResNet-50 features https://github.com/jindongwang/transferlearning/blob/master/data/dataset.md.

**Imbalance settings**   We consider different class-ratios between domains to simulate label-shift and denote with a "s" prefix, the subsampled datasets. For Digits, we explicitly provide the class-ratios as Rakotomamonjy et al. (2021) (e.g. for high imbalance, class 2 accounts for the 7% of target samples while class 4 accounts for 22% of target samples). For Visda12, Office31 and OfficeHome, subsampled datasets only consider a small percentage of source samples for the first half classes as Combes et al. (2020) (e.g. Office31 considers 30% of source samples in classes below 15 and uses all source samples from other classes and all target samples).

Table 8: Label imbalance settings

| Dataset | Configuration | $\boldsymbol{p}_S^Y$ | $\boldsymbol{p}_T^Y$ |
|---|---|---|---|
| Digits | balanced | $\{\frac{1}{10}, \cdots, \frac{1}{10}\}$ | $\{\frac{1}{10}, \cdots, \frac{1}{10}\}$ |
|  | subsampled mild | $\{\frac{1}{10}, \cdots, \frac{1}{10}\}$ | $\{0,1,2,3,6\} = 0.06, \{4,5\} = 0.2, \{7,8,9\} = 0.1$ |
|  | subsampled high | $\{\frac{1}{10}, \cdots, \frac{1}{10}\}$ | $\{0,1,2,3,6,7,8,9\} = 0.07, \{4,5\} = 0.22$ |
| VisDA12 | original | $\{0 - 11\} : 100\%$ | $\{0 - 11\} : 100\%$ |
|  | subsampled | $\{0 - 5\} : 30\% \ \{5 - 11\} : 100\%$ | $\{0 - 11\} : 100\%$ |
| Office31 | subsampled | $\{0 - 15\} : 30\% \ \{15 - 31\} : 100\%$ | $\{0 - 31\} : 100\%$ |
| OfficeHome | subsampled | $\{0 - 32\} : 30\% \ \{33 - 64\} : 100\%$ | $\{0 - 64\} : 100\%$ |

**Hyperparameters** Domain-invariant methods weight alignment over classification; we tuned the corresponding hyperparameter for $\texttt{WD}_{\beta=0}$ in the range $[10^{-4}, 10^{-2}]$ and used the one that achieves the best performance on other models. We also tuned $\lambda_{OT}$ in problem (CAL) and fixed it to $10^{-2}$ on Digits and $10^{-5}$ on VisDA12, Office31 and OfficeHome. Batch size is $N_b = 200$ and all models are trained using Adam with learning rate tuned in the range $[10^{-4}, 10^{-3}]$. We initialize NN for classifiers and feature extractors with a normal prior with zero mean and gain 0.02 and $\phi$ with orthogonal initialization with gain 0.02.

**Training procedure** We fix $N_e$ the number of epochs to 50 on Digits, 150 on VisDA12 and 100 on Office31, OfficeHome; OSTAR requires smaller $N_e$ to converge. Critic iterations are fixed to $N_v = 5$ which worked best for all baseline models; for OSTAR higher values performed better. For all models, we initialize $f_S, g$ for 10 epochs with (3). Then, we perform alignment either through domain-invariance or with a domain-mapping until we reach the total number of epochs. GeTarS models IW-WD, MARSc, MARSg, OSTAR perform reweighting with estimates refreshed every $N_u = 2$ epochs in the first 10 alignment epochs, every $N_u = 5$ epochs after. OSTAR minimizes (CAL) + (SS) for 10 epochs on Digits, Office31, OfficeHome and 5 epochs on VisDA12, then minimizes (CAL) + (SSg) for remaining epochs.

**Architectures** For Digits, our feature extractor $g$ is composed of three convolutional layers with respectively 64, 64, 128 filters of size $5 \times 5$ interleaved with batch norm, max-pooling and ReLU. Our classifiers $(f_S, f_N)$ are three-layered fully-connected networks with 100 units interleaved with batch norm, ReLU. Our discriminators are three-layered NN with 100 units and ReLU activation. For VisDA12 and Office31, OfficeHome, we consider pre-trained 2048 features obtained from a ResNet-50 followed by 2 fully-connected networks with ReLU and 100 units for VisDA12, 256 units for Office31, OfficeHome. Discriminators are 2-layer fully-connected networks with respectively 100/1 units on VisDA12, 256/1 units on Office31, OfficeHome interleaved with ReLU. Classifiers are 2-layer fully-connected networks with 100/$K$ units on VisDA12, single layer fully-connected network with $K$ units on Office31, OfficeHome. $\phi$ is a ResNet with 10 blocks of two fully-connected layers with ReLU and batch-norm.

**Implementation of target proportion estimators** OSTAR and IW-WD use the confusion based estimator in Lipton et al. (2018) and solve a convex optimization problem ((4) in Combes et al. (2020) and CAL w.r.t $\boldsymbol{p}_N^Y$ for OSTAR) which has an unique solution if the soft confusion matrix $\mathbf{C}$ is of full rank. We implement the same optimization problem using the parallel proximal method from Pustelnik et al. (2011) instead of cvxopt[2] used in Combes et al. (2020). MARSc and MARSg (Rakotomamonjy et al., 2021) use linear programming with POT[3] to estimate proportions with optimal assignment between conditional distributions. Target conditionals are obtained with hierarchical clustering or with a Gaussian Mixture Model (GMM) using sklearn[4]. MARSg has some computational overhead due to the GMM.

---

[2]http://cvxopt.org/
[3]https://pythonot.github.io/
[4]https://scikit-learn.org/

