# OpenReview forum: "Mapping conditional distributions for domain adaptation under generalized target shift"
_ICLR.cc/2022/Conference — ICLR 2022 Poster_

### Official Review · Reviewer_9ZXq · 2021-10-31

**Correctness:** 4
**Technical Novelty And Significance:** 3
**Empirical Novelty And Significance:** 2
**Recommendation:** 5
**Confidence:** 2

**Main Review:**

Strength:
1. The paper is well-motivated from the Generalized Target Shift setup and uses optimal transport to map the source distribution to the target.
2. The proposed approach is supported by theoretical analysis that minimizes the upper bound of the target risk.
3. The proposed approach demonstrates sota results over the baselines on three datasets.

Weaknesses:
1. The theory and implementation are disconnected with the use of information maximization. The main argument is the use of optimal transport for joint alignment and classification, and it is not clear why it requires information maximization if the joint alignment and classification are successful. Because the information maximization (IM) approach is proposed in the context of source-free transfer, it is not clear whether the use of IM hampers the learning of optimal transport.
2. In the experiments, the IM component in SS and SSg substantially improves (+9%) the results of joint alignment and classification, while the main focus of this paper is joint alignment and classification. But in the main empirical results in Tab. 1, only results of OSTAR are presented. It will be more informative and convincing to also include the results of CAL alone to help fully understand the impact of optimal transport.

**Summary Of The Paper:**

The paper proposes an approach for Generalized Target Shift (GeTarS) where both conditional and label shift are present in the target domain. The proposed approach, Optimal Sample Transformation and Reweight (OSTAR), uses optimal transport to transform the latent space and uses information maximization to refine the classifier's decision boundaries. Theoretical analysis shows the proposed approach minimizes the the components in the target risk's upper bound. Empirical studies are carried out on three datasets and ablation studies were carried out to analyze the impact of MI.

**Summary Of The Review:**

The idea of using optimal transport to address the problem of generalized target shift is interesting, but the main argument of this paper, optimal transport and its theoretical analysis, is not sufficiently evaluated and is confounded by the additional component information maximization. Based on the empirical studies, it appears information maximization has substantially more contribution than optimal transport, and the outcome of optimal-transport (CAL) is missing in the main results (Tab. 1).

---

> ### Author Response · Authors · 2021-11-18
> **Answers to Reviewer 9ZXq - Part 1/2**
>
> We tried to make a synthetic response and we will be happy to discuss our answers in more details should some points remain unclear.
>
> *Recap* As stated by the reviewer, our main contribution is to define a theoretically grounded alignment method for GeTarS based on optimal transport (OT) and acting on pretrained representations. We found that the performance of OSTAR depends on the discriminativity of target representations which are fixed. We thus proposed a simple extension based on Information Maximization (IM) which updates the encoder to improve target discriminativity. IM was originally introduced for source-free adaptation without alignment (Liang et al. (2020)). In this context, target predictions are prone to errors and there is no principled way to mitigate this problem. In our case, we have access to the source samples and can define and minimize an upper-bound to target errors in (2) with OSTAR. We explain in the following that IM combined with alignment better reduces this upper-bound.
>
> 1)
> *"It is not clear why IM is required if joint alignment and classification is successful"*
> With objective function (CAL), OSTAR minimises the upper-bound of the target error at fixed representations in eq (2). (C) corresponds to classification in the representation space $\mathcal{Z}$ and (A) and (L) to alignment in $\mathcal{Z}$. If joint alignment and classification is successful (i.e. terms (A), (L) and (C) are close to zero) then IM is indeed useless. However, since $g$ has been trained with source labels, nothing guarantees that term (C) at optimum (when (A) and (L) are zero) is low. In practise, this is usually not the case. To reduce this term, one can act upon the encoder $g$. Assuming that the domains are perfectly aligned, Proposition 1 shows that $p_N(Z,Y)=p_T(Z,Y)$ where $N$ is labelled while $T$ is unlabelled. The problem then collapses to a standard semi-supervised problem without domain shift and various methods can be used to reduce the target error w.r.t. the encoder $g$. An option is to use pseudo-labels obtained by target predictions of the classifier. An alternative is to use conditional entropy defined in (7). The two are equivalent. We consider in practise an extension of conditional entropy, IM.
>
> *"Theory is disconnected with the use of IM" and "It is not clear if IM hampers the learning of OT"*
> When the optimal alignment is not reached, we believe that our practice is still in accordance with our theory. Indeed, the theory provides an upper-bound of the target error at fixed $g$, $\phi$ and $f_N$ in eq (2). Hence, it provides a guideline for choosing between two triplets of ($g$, $\phi$, $f_N$), but it does not tell us how to optimally improve on each term of this triplet. The analysis at optimum in the previous paragraph illustrated the usefulness of optimizing $g$ in a semi-supervised setting using target predictions. Therefore, we propose a two-stage training process where representations are first aligned, then alignment is combined with IM. In practise, we show through new experimental results that, in a noisy and imperfect situation, IM helps to better optimize these three functions. This is witnessed in Table 1 below, which compares the values of terms (C), (A) and (L) from the upper-bound (2) with IM (columns OSTAR+IM) and without IM (columns OSTAR). Remember that term (C) is related to classification and terms (A) and (L) to alignment. The value of the upper-bound is smaller with IM, in particular the value of (A) and (L); this shows that *IM does not hamper alignment but facilitates it*. Terms (A) and (L) are computed with the primal formulation of OT using POT https://pythonot.github.io. Note that target conditionals are assumed to be known here so that term (L) can be computed. Term (C) is easily computed using labelled samples in domain $N$. We include this analysis and table in the revision (Table 6 in the revision).

---

> > ### Author Response · Authors · 2021-11-18
> > **Answers to Reviewer 9ZXq - Part 2/2**
> >
> >
> > Table 1. Value after 50 training epochs of term (A) -$W_1(p_N^{\phi}(Z), p_T(Z))$- ($\downarrow$), term (L) -$W_1(\sum_{k=1}^{K} p_{N}^{Y=k} p_T(Z|Y=k), p_T(Z))$- ($\downarrow$) and term (C) -$\epsilon_N^g(f_N)$- ($\downarrow$) without and with IM in OSTAR. Best results are in **bold**.
> >
> > |          | Term | (A) | Term    | (L)            | Term  | (C)|
> > |----------|----------|----------|------------------------|----------|----------|----------|
> > | Setting  | OSTAR    | OSTAR+IM | OSTAR                  | OSTAR+IM | OSTAR    | OSTAR+IM |
> > |          |          |          | MNIST|   $\rightarrow$       |    USPS       |          |
> > | Balanced | $49.83$ | $\textbf{16.56}$ | $1.45$ | $\mathbf{0.18}$ | $2.19 \times 10^{-3}$ | $\mathbf{0.918 \times 10^{-3}}$|
> > | Mild     | $39.31$ | $\textbf{19.13}$ | $10.78$ | $\mathbf{0.24}$ | $1.65 \times 10^{-3}$ | $\mathbf{0.931 \times 10^{-3}}$|
> > | High    | $38.60$ | $\textbf{21.10}$ | $12.78$ | $\mathbf{0.74}$ | $1.76\times 10^{-3}$ | $\mathbf{0.510 \times 10^{-3}}$|
> > |          |          |          | USPS |  $\rightarrow$ |    MNIST             |          |
> > | Balanced  | $235.43$ | $\textbf{86.65}$ | $7.08$ | $\mathbf{0.52}$ | $4.46 \times 10^{-3}$ | $\mathbf{0.495 \times 10^{-3}}$|
> > | Mild   | $188.67$ | $\textbf{104.66}$ | $20.99$ | $\mathbf{1.05}$ | $3.98 \times 10^{-3}$ | $\mathbf{0.399 \times 10^{-3}}$|
> > | High  | $181.64$ | $\textbf{123.83}$ | $21.62$ | $\mathbf{0.82}$ | $4.51\times 10^{-3}$ | $\mathbf{0.616 \times 10^{-3}}$|
> >
> > 2.  *"It is more informative to include the results without IM"*
> > As requested, we included the results of OSTAR without IM i.e. (CAL) in our main Table 1 in the revision. As requested by reviewer R-cdrt, we averaged results over all adaptation problems and imbalance settings in a same dataset. We report in Table 2 below part of these results. We stress that the results of OSTAR without IM depend on the discriminativity of pretrained target representations, while domain-invariant method are less prone to this problem as they update the encoder i.e. they change target representations. Interestingly, OSTAR without IM is already competitive compared to IW-WD on VisDA, Office31 and OfficeHome and to MARSg on Digits although it keeps target representations fixed unlike these two methods. OSTAR+IM leads to practical gains on all datasets as it makes our model less prone to initial lack of target discriminativity.
> >
> > Table 2. Balanced accuracy ($\uparrow$) over 10 runs. The best performing model is indicated in **bold**. Results are aggregated over all imbalance scenarios and adaptation problems within a same dataset.
> >
> > |Setting|Source|MARSg|MARSc|IW-WD|OSTAR|OSTAR+IM|
> > |---|---|---|---|---|---|---|
> > |||| Digits ||||
> > |balanced|$74.98 \pm 3.8 $|$92.18 \pm 2.2 $|$94.91 \pm 1.4$|$95.89 \pm 0.5$|$91.66 \pm 0.9$|$\mathbf{97.51 \pm 0.3}$ |
> > |aver. subsampled|$75.05 \pm 3.1$|$91.87 \pm 2.0$|$93.75 \pm 1.4$|$93.22 \pm 1.1$|$88.39 \pm 1.5$|$\mathbf{96.69 \pm 0.7} $ |
> > ||||VisDA12||||
> > |original|$48.63 \pm 1.0$|$55.62 \pm 1.6$|$55.33 \pm 0.8$|$51.88 \pm 1.6$|$50.37 \pm 0.6$|$\mathbf{59.24 \pm 0.5}$ |
> > |aver. subsampled|$42.46 \pm 1.4$|$55.00 \pm 1.9$|$51.86 \pm 2.0$|$50.65 \pm 1.5$|$49.05 \pm 0.9$|$\mathbf{58.84 \pm 1.0}$ |
> > ||||Office31||||
> > |aver. subsampled|$74.50 \pm 0.5$|$80.20 \pm 0.4$|$80.00 \pm 0.5$|$77.28 \pm 0.4$|$76.19 \pm 0.8$|$\mathbf{82.61 \pm 0.4}$ |
> > ||||OfficeHome||||
> > |aver. subsampled|$50.56 \pm 2.8$|$ 56.60 \pm 0.4$|$56.22 \pm 0.6$|$ 54.87 \pm 0.4$|$54.64 \pm 0.7$|$\mathbf{59.51 \pm 0.4}$|
> >
> > **References in this answer**
> >
> > Liang et al. Do We Really Need to Access the Source Data? {S}ource Hypothesis Transfer for Unsupervised Domain Adaptation. ICML 2020.

---

> > > ### Comment · Reviewer_9ZXq · 2021-11-19
> > > **Additional questions on IM, baselines**
> > >
> > > I'd like to thank the authors for their detailed responses.
> > >
> > > In the new Table 2 from the author response, I'm concerned that OSTAR does not outperform all the baselines, except the source, without using IM. Is it fair to compare OSTAR+IM with IW-WD, rather than with IW-WD+MI? It is possible that MI can improve all the baselines and unclear whether the contribution of OSTAR+IM is substantial.

---

> > > > ### Author Response · Authors · 2021-11-20
> > > > **Clarifications on baselines+IM**
> > > >
> > > > We thank the reviewer for his/her reply. Here are some clarifications.
> > > >
> > > > *"I'm concerned that OSTAR does not outperform all the baselines, except the source, without using IM"*
> > > > Again, we stress that *OSTAR does not change representations while these baselines do*. Thus, the comparison is not fair between domain-invariant baselines and OSTAR. Yet, despite not changing representations, OSTAR already achieves comparable results especially w.r.t to IW-WD on VisDA, Office31 and OfficeHome and w.r.t. MARSg on Digits.
> > > >
> > > > *"It is possible that IM improves all the baselines and unclear whether the contribution of OSTAR+IM is substantial"* We actually already evaluated the effect of adding IM to two of our Generalized Target Shift baselines (MARSc and IW-WD) on several of our adaptation problems in Appendix Table 5 (with corresponding text p8 of our revision) even if this was not part of their original work. We present a copy of these results in Table 3 below. We observe that IM improves the performance of those baselines on VisDA, Office31 and OfficeHome, while on Digits, there are important performance degradations on MNIST$\rightarrow$USPS on the high imbalance scenario. The performance remains however below the ones of OSTAR+IM on all datasets.
> > > >
> > > > The two families of methods have different properties. Domain-invariant baselines already optimize target representations through alignment, while OSTAR does not. This is an advantage for domain-invariant baselines since it removes the dependency w.r.t. initial representations. Hence, since these baselines already adjust the representations, IM does not bring much to these methods. OSTAR, like any method that learns a mapping between non domain-invariant source and target representations, is highly dependent on the initial representations learned via the encoder. This is why, for this family of methods, there should be another mechanism (IM here) that improves the initial target representations. Overall, the performance of OSTAR+IM makes it a competitive alternative to state-of-the-art domain-invariant methods even when IM is added to these baselines.
> > > >
> > > > Table 3. Effect of adding IM to MARSc, IW-WD and OSTAR on balanced accuracy ($\uparrow$). Best results are in **bold**.
> > > >
> > > > |Setting / Model | MARSc | MARSc+IM | IW-WD | IW-WD+IM | OSTAR | OSTAR+IM|
> > > > |-|-|-|-|-|-|-|
> > > > |||MNIST|$\rightarrow$|USPS|||
> > > > |balanced | $96.44 \pm 0.3$ | $\mathbf{97.92 \pm 0.2}$ | $96.10 \pm 0.3$ | $\mathbf{97.91 \pm 0.1}$ | $95.12 \pm 0.6$  | $ 96.91 \pm 0.3 $|
> > > > |mild imbalance| $95.18 \pm 0.9 $ | $95.47 \pm 1.1$ | $94.72 \pm 0.4$ | $95.74 \pm 0.6$ | $91.77 \pm 1.2$ | $\mathbf{96.18 \pm 1.0} $|
> > > > |high imbalance| $ 95.07 \pm 0.6 $ |$93.76 \pm 0.5$ | $94.60 \pm 0.8$ | $91.73 \pm 0.6$ | $88.55 \pm 1.1$ | $\mathbf{96.06 \pm 0.6} $|
> > > > |||USPS|$\rightarrow$|MNIST|||
> > > > |balanced | $ 93.37 \pm 2.5$ | $93.03 \pm 1.9$ | $ 95.68 \pm 0.6$ | $96.17 \pm 0.5$ | $88.19 \pm 1.1$ | $\mathbf{98.11 \pm 0.2} $|
> > > > |mild imbalance| $93.20 \pm 2.8 $ | $94.60 \pm 1.7$ | $92.73 \pm 1.5$  | $92.65 \pm 1.0$ | $88.34 \pm 1.3$ | $\mathbf{97.44 \pm 0.5}$ |
> > > > |high imbalance| $91.54 \pm 0.9 $ | $90.16 \pm 2.0$ | $90.81 \pm 1.5$ | $91.26 \pm 1.1$ | $84.87 \pm 2.3$ | $\mathbf{97.08 \pm 0.6}$ |
> > > > ||||VisDA12||||
> > > > | VisDA | $55.33 \pm 0.8$ | $57.57 \pm 0.8$ | $51.88 \pm 1.6$ | $57.63 \pm 0.1$ | $50.37 \pm 0.6$ | $\mathbf{59.24 \pm 0.5}$|
> > > > |sVisDA | $51.86 \pm 2.0$ | $57.06 \pm 0.8$ | $50.65 \pm 1.5$ | $57.62 \pm 0.7$ | $49.05 \pm 0.9$ | $\mathbf{58.84 \pm 1.0}$|
> > > > ||||Office31||||
> > > > |sW-A | $63.80 \pm 0.3$ | $68.12 \pm 0.5$ | $60.25 \pm 0.2$ | $67.42 \pm 0.8$  | $59.62 \pm 0.6$ | $\mathbf{69.99 \pm 0.1}$|
> > > > |sA-W | $81.05 \pm 0.7$ | $81.83 \pm 1.9$ | $75.84 \pm 0.7$ | $82.34 \pm 1.6$ | $75.30 \pm 1.0$ | $\mathbf{83.91 \pm 0.5}$|
> > > > ||||OfficeHome||||
> > > > |sR-P | $72.72 \pm 1.1$ | $\mathbf{75.17 \pm 0.6}$ | $72.90 \pm 0.7$ | $74.94 \pm 0.6$ | $71.77 \pm 0.4$ | $\mathbf{75.58 \pm 0.6}$|
> > > > |sR-A | $53.37 \pm 1.3$ | $54.20 \pm 1.3$ | $53.44 \pm 0.6$ | $54.50 \pm 1.1$  | $55.15 \pm 0.8$ | $\mathbf{56.28 \pm 0.5}$|
> > > > |sR-C | $45.30 \pm 1.2$ | $48.17 \pm 1.2$ | $42.66 \pm 1.0$ | $47.93 \pm 1.7$ | $43.02 \pm 3.1$ | $\mathbf{49.07 \pm 0.9}$|

---

> > > > > ### Author Response · Authors · 2021-11-25
> > > > >
> > > > > As the discussion period ends soon, we hope that our answer has clarified reviewer 9ZXq's remaining question. We are happy to provide any additional clarifications if still needed.

---

### Official Review · Reviewer_cdrt · 2021-11-01

**Correctness:** 3
**Technical Novelty And Significance:** 3
**Empirical Novelty And Significance:** 3
**Recommendation:** 6
**Confidence:** 3

**Main Review:**

Pros:
(1)	The proposed OSTAR well handles the GeTarS case and improves the target discriminativity.
(2)	The paper gives both the theoretical and empirical analyses for the proposed OSTAR.
(3)	The experiments show improvements over existing baselines.

Here are some comments/questions for improvements:
(1)	In introduction, the paper gives two limitations of existing works, discriminativity and strong assumptions. Please clarify (1) what’s the advantage of OSTAR compared with those UDA methods that explicitly preserve the data discriminativity in domain-invariant feature learning, e.g., [ref1] and [ref2]; and (2) please clearly state the strong assumptions or indicates the related works.

[ref1] Structure preservation and distribution alignment in discriminative transfer subspace learning

[ref2] Discriminative Transfer Feature and Label Consistency for Cross-Domain Image Classification

(2)	 As $g$ is learned by eq (3) and fixed afterwards. It is unclear how to guarantee the target discriminativity is preserved when learning $g$.

(3)	Regarding eq (OT), please also give reference for “least action principle measured by monge transport cost”.

(4)	According to eq (9), the constraints in eq (OT) should be $p_N^\phi(Z) = p_S(Z)$? If so, It is unclear how to link $p_T(Z)$ with $p_N^\phi(Z)$.

(5)	I understand that each of assumption 1-4 is less restrictive than the corresponding one in the related work. However, proposition 1 requires $Z$ satisfying all the four assumptions. In this sense, it is unclear how significant the proposition 1 is in terms of the flexibility.

(6)	Some important references are missing, e.g., [ref3] and [ref4].

[ref3] Flexible transfer learning under support and model shift

[ref4] Generalization Bounds for Transfer Learning under Model Shift

(7)	In the overall objective, why only $\mathcal{L}^g_{OT}$ is weighted by an importance factor? How about the term $\mathcal{L}^g_{wd}$ is also weighted? Similarly, for each term in eqs. (SS) and (SSg), how about they are weighted?

(8)	Regarding $\lambda_{OT}$, the authors only conduct 0 and 0.01 cases. How about the other values? More general, It is interesting to see the sensitivity analyses on such an importance factor.

(9)	For table 1, it is better to show the average results for each dataset.


**Summary Of The Paper:**

In this paper, the authors unsupervised domain adaptation (UDA) where both the label conditional and marginal distributions are different between the source and target domains (GeTarS). The authors propose an approach, OSTAR, to align pretrained representations under GeTarS. A new theoretical bound under some assumptions is derived. Experimental studies also show the effectiveness of OSTAR.

**Summary Of The Review:**

Overall, the paper is well-written and makes some contributions. For more detailed comments, see above.

---

> ### Author Response · Authors · 2021-11-18
> **Answers to Reviewer cdrt - Part 1/3**
>
> *1) a) What are the advantages of OSTAR over discriminativity-preserving domain-invariant methods ?*
> We thank the reviewer for the references; we missed them despite a thorough bibliographic search. To summarize, the authors of these publications proposed to preserve target discriminativity while learning invariant representations either with graph regularization and low-rank / sparse constraint on a reconstruction coefficient matrix in SPDA (Xiao et al. (2019)) or a variant of the triplet loss in DTLC (Li et al. (2020)). This is similar to the idea we used in OSTAR for improving target discriminativity with Information Maximization (IM). OSTAR has the following main differences over these approaches. First, OSTAR tackles the more general GeTarS setting while the other methods only deal with covariate shift. The ideas in SPDA and DTLC for improving discriminativity could probably be adapted to the GeTarS setting, but this was not done before to our knowledge. Second, OSTAR guarantees controlled target prediction errors with our objective function, which allows IM to behave well. The other methods usually rely on heuristics, which can be brittle, to avoid performance degradation when improving discriminativity (e.g. DTLC selects relevant pseudo-labels with a heuristic). A third difference is that OSTAR fixes the encoder during alignment while these other methods align through the encoder via domain-invariance. An advantage of fixing representations is that this process is less prone to the well-known instability of the adversarial alignment methods in most domain invariant approaches. This is particularly relevant on domains where there is no well established DL standard architectures such as click-through-rate prediction, spam filtering etc. in contrast to domains such as vision where this problem was well studied. Moreover, OSTAR introduced native OT regularization biases, which do not exist in domain-invariant methods, to better control alignment and improve stability. Finally, OSTAR is a fully deep UDA method while some alternative e.g. SPDA and DTLC have linear components.
>
> *1) b) Clarify where the strong assumptions in existing work are mentioned*
> We added a sentence in the revision, which mentions that the strong assumptions used in related work are further described in Section 2.3. The main assumptions are: in Zhang et al. (2013), Gong et al. (2016), the ACons assumption which states the existence of a map matching the right conditional pairs i.e. $\forall k~\phi$#$(p_{S}(Z|Y=k))=p_T(Z|Y=k)$ and the A2Cons assumption which states linear independence of linear combinations of source and target conditionals, in $\mathcal{X}$ respectively $\mathcal{Z}$. For Combes et al. (2020), the GLS assumption which imposes that $p_{S}(Z|Y)=p_T(Z|Y)$. Finally for both Combes et al. (2020), Rakotomamonjy et al. (2020), the cluster assumption on target samples which may not hold.
>
> *2) How to guarantee preservation of target discriminativity ?*
> Neither domain-invariant works in Xiao et al. (2019), Li et al. (2020) nor OSTAR guarantee that the discriminativity is preserved in the target space. A standard approach is to use semi-supervised heuristics to better enforce it. After aligning representations with our OT alignmnent problem (OT), OSTAR applies a semi-supervised refinement step (SSg) explained in Section 4 based on IM. This step updates the encoder $g$ and plays the same role as triplet loss in Li et al. (2020). IM involves two terms; the first is conditional entropy on target predictions in (7) which, as noted by Lee et al. (2013), is equivalent to using target pseudo-labels in a classification objective and the second is a diversity term in (8) which accounts for noise in target predictions, when the optimum is not reached. OSTAR controls the error of target predictions by minimizing an upper-bound in (2), which allows IM to behave properly under GeTarS.
>
> *3) Missing reference for OT and least action principle*
> We added a citation to Santambrogio (2015) in the revision.
>
> *4) Clarify the constraint in (OT)*
> Our OT problem in (OT) aims at mapping source conditionals to target ones with $\phi$ such that the transport cost of source conditionals is minimal. As target conditionals are unknown, our proxy alignment problem matches the marginal of domain $N$ defined by transporting and reweighting source conditionals with $(\phi, p_N^Y)$ and the target marginal $p_T(Z)$ i.e. $p_N^\phi(Z)=p_T(Z)$. This adjusts the standard Monge OT problem in (9), where $\alpha=p_S(Z)$ and $\beta=p_T(Z)$, to the GeTarS UDA problem. We provide more details on the differences between (OT) and (9) in the answer "Discrepancy between objective (OT) and the Monge OT problem (9)" to R-FHUc.

---

> > ### Author Response · Authors · 2021-11-18
> > **Answers to Reviewer cdrt - Part 2/3**
> >
> > *5) How flexible is Proposition 1 ?*
> > All the methods providing theoretical guarantees rely on a series of assumptions, either explicit or implicit. In this regard, our assumptions all together are not more restrictive than the ones used in these papers. A contribution of our paper is that it provides stronger guarantees than e.g. some GeTarS work as Combes et al. (2020) which does not guarantee recovering $p_T^Y$. Assumptions 1 and 4 are common to several papers. Assumption 1 is easily met in practice since it could be forced by training a classifier on the source labels. Assumption 4 is said to be met in high dimensions (Redko et al. (2019); Garg et al. (2020)). Assumption 3 says that source and target clusters from the same class are globally (there is a sum in the condition) closer one another than clusters from two different classes. This assumption is required to cope with the absence of target labels. Because of the sum, it could be considered as a reasonable hypothesis. It is milder than the hypotheses made in papers offering guarantees similar to ours (Zhang et al. (2013); Gong et al. (2016)). Assumption 2 is new to this paper. It guarantees that mass of a source conditional will be entirely transferred to a target conditional. This property is key to show that OSTAR matches source conditionals and their corresponding target conditionals, which is otherwise very difficult to guarantee under GeTarS as both target conditionals and proportions are unknown. In practise, our optimization problem in (CAL) mitigates this problem by learning how to reweight source conditionals to improve alignment. To better verify that mass is preserved across clusters, we could additionally apply Laplacian regularization (Ferradans et al. (2013), Carreira-Perpinan et al. (2014)) into our OT map $\phi$ but this was not developed in the paper. Thus Proposition 1 applies *in theory*. *In practise*, we agree that there is no reason for the assumptions to be verified altogether. Moreover, it might be difficult to perfectly align representations to achieve the optimum of problem (OT). The experimental evaluation however demonstrates that our method optimizes a criterion which controls to a good extent the target risk. Moreover our t-SNE visualization in Appendix A show that representations are reasonably aligned with OSTAR.
> >
> > *6) Missing references on UDA and model shift*
> > We included the two additional references in the introduction of our revision to account for the literature on model shift as an alternative to GeTarS.
> >
> > *7) Which terms are weighted in the objective function ?*
> > $\lambda_{OT}L_{OT}^g + L_{wd}^g$ i.e. (5)+(6) in (CAL) is the Lagrangian relaxation of our constrained optimization problem in (OT), thus we do not additionally weight $L_{wd}^g$ (6). In (SS) and (SSg) no terms are weighted as it did not lead to significant improvements. The classification loss $L_c^g(f_N, N) + L_{ent}^g(f_N, T) + L_{div}^g(f_N, T) + L_{c}^g(f_S, S)$ i.e. (4)+(7)+(8)+(3) in (SSg) could be weighted w.r.t. the alignment terms $\lambda_{OT}L_{OT}^g + L_{wd}^g$ i.e. (5)+(6) in (CAL); yet, in practise, this yields small gains. To avoid further complexity in hyperparameter tuning, we do not perform other weighting than $\lambda_{OT}$.
> >
> > *8) Test additional values of $\lambda_{OT}$ in the ablation study in Table 7 in the revision*
> > We report in Table 1 below, balanced accuracy results ($\uparrow$) when $\lambda_{OT} \in $ {$10^{-3}, 10^{-2}, 10^{-1}, 1, 10^{4}$} on MNIST$\rightarrow$USPS under various imbalance settings. For the full study please refer to Table 7 in the revision. We observe that for high values of $\lambda_{OT}$ e.g. $\lambda_{OT}=10^{4}$, OSTAR retrieves the performance of the Source model which uses only labelled source samples without adaptation. Indeed, as the cost of transporting source conditionals is high for high values of $\lambda_{OT}$, $\phi$ is constrained to be close to identity. We find an optimal $\lambda_{OT}$ between this edge case and $\lambda_{OT}=0$, when no transport is applied.
> >
> > Table 1. Detailed analysis of the impact of $\lambda_{OT}$ on balanced accuracy ($\uparrow$) on MNIST$\rightarrow$USPS with fixed initialization gain at 0.02. Best results are in **bold**.
> >
> > |Shift / $\lambda_{OT}$   | $\lambda_{OT}=0$   | $\lambda_{OT}=10^{-3}$   | $\lambda_{OT}=10^{-2}$   | $\lambda_{OT}=10^{-1}$   | $\lambda_{OT}=1$   | $\lambda_{OT}=10^4$   | Source   |
> > |---|---|---|---|---|---|---|---|
> > |balanced   | $94.92 \pm 0.6$   | $\mathbf{96.02 \pm 0.2}$   | $95.12 \pm 0.6$   | $89.76 \pm 1.2$   | $91.95 \pm 1.2$   | $87.03 \pm 1.8$   | $86.02 \pm 1.4$   |
> > |mild   | $88.28 \pm 1.5$   | $88.63 \pm 1.3$   | $\mathbf{91.77 \pm 1.2}$   | $90.17 \pm 1.7$  | $88.51 \pm 1.2$   | $88.42 \pm 1.6$   | $89.08 \pm 0.5$   |
> > |high   | $85.24 \pm 1.6$   | $85.38 \pm 1.4$   | $88.55 \pm 1.1$   | $\mathbf{89.10 \pm 1.2}$   | $\mathbf{88.82 \pm 1.1}$   | $86.94 \pm 1.1$   | $86.73 \pm 1.9$   |

---

> > > ### Author Response · Authors · 2021-11-18
> > > **Answers to Reviewer cdrt - Part 3/3**
> > >
> > > *9) "It is better to average results for each dataset"*
> > > Thank you for this advise which we applied to the revision.
> > >
> > > **References in this answer**
> > >
> > > Ting Xiao, Peng Liu, Wei Zhao, Hongwei Liu, and Xianglong Tang.   Structure preservation anddistribution alignment in discriminative transfer subspace learning.Neurocomputing, 337:218–234, 2019
> > >
> > > Shuang Li, Chi Harold Liu, Limin Su, Binhui Xie, Zhengming Ding, C. L. Philip Chen, and DapengWu.  Discriminative transfer feature and label consistency for cross-domain image classification.IEEE Transactions on Neural Networks and Learning Systems, 31(11):4842–4856, 2020.
> > >
> > > Lee, D.-H. Pseudo-label: The simple and efficient semi- supervised learning method for deep neural networks. In Workshop on challenges in representation learning, ICML, 2013.
> > >
> > > F. Santambrogio. Optimal transport for Applied Mathematicians. Birkhäuser, 2015.
> > >
> > > Rémi Tachet des Combes, Han Zhao, Yu-Xiang Wang, and Geoff Gordon.  Domain adaptation with conditional distribution matching and generalized label shift. NeurIPS, 2020.
> > >
> > > Ievgen Redko, Nicolas Courty, Réemi Flamary, and Devis Tuia. Optimal transport for multi-source domain adaptation under target shift. AISTATS 2019.
> > >
> > > Saurabh Garg, Yifan Wu, Sivaraman Balakrishnan, and Zachary Lipton.   A unified view of labelshift estimation. NeurIPS 2020.
> > >
> > > Kun Zhang, Bernhard Schölkopf, Krikamol Muandet, and Zhikun Wang.  Domain adaptation under target and conditional  shift. ICML 2013
> > >
> > > Mingming  Gong,  Kun  Zhang,  Tongliang  Liu,  Dacheng  Tao,  Clark  Glymour, and  Bernhard Scholkopf. Domain  adaptation  with  conditional  transferable  components. ICML 2016.
> > >
> > > S. Ferradans, N. Papadakis, J. Rabin, G. Peyré, and J.-F. Aujol, “Regularized discrete optimal transport,” in Scale Space and Variational Methods in Computer Vision, SSVM, 2013, pp. 428–439.
> > >
> > > M. Carreira-Perpinan and W. Wang, “LASS: A simple assignment model with laplacian smoothing,” in AAAI Conference on Artificial Intelligence, 2014.

---

### Official Review · Reviewer_FHUc · 2021-11-04

**Correctness:** 4
**Technical Novelty And Significance:** 3
**Empirical Novelty And Significance:** 2
**Recommendation:** 8
**Confidence:** 4

**Main Review:**

Strengths:
* The core of the approach is well motivated and theoretically appealing
* Unlike many other deep learning DA papers, this one clearly puts forward assumptions and bases theoretical results on them
* The domain generalization bound provides interesting insights regarding class proportion and imbalance
* The experimental section is quite thorough, provides meaningful error estimates, various ablations, and achieves impressive SOTA results
* The paper is overall very well written, and is quite fun to read

Weaknesses:
* Although the risk bound motivates Wasserstein distance minimization between latent marginals and class-weighted conditional distributions, the actual optimization objective used combines a plethora of terms, so it's not clear how relevant the theoertical results are for the actual method proposed
* There is not (theoretical or empirical) runtime/complexity analysis
* The intuition/motivation for some of the assumptions could use a bit more detail. Some of them are taken ipso facto from prior work, without much justification. It would be useful to have it here (even if in the appendix). This is especially true for Assumption 2, as per question below.
* One aspect that is not discussed enough is the effect of the latent encoder g in the overall performance of the method, e.g., what happens if g is not rich enough (e.g., initial layers of a NNEt $f\circ g(x)$, for which $g$ is shallow and $f$ is deep). Moreover, given two possible encoders g1, g2, how should one choose between them? is there a way to easily verify which one leads to the 4 Assumptions being satisfied? Should one try to estiamte the bound for both of these and choose the one with the tighter one?

Questions/Comments/Suggestions:
* Objective OT is *almost* the Monge OT problem between $p_T(Z)$ and $p_S(Z)$ (by moving sum inside and using law of total prob), except for the fact that the pushfoward constraint is not exactly $\phi_{\sharp}
p_S(Z) = p_T(Z)$, but rather one that uses the class proportions $p_N$ rather than $p_S$. I was hoping to get more clarity on the reason for this discrepancy in the paper or Appendix, but it seems it isn't really discussed anywhere.
* Assumption 2 feels potentially very restrictive: it says that the $\phi$ must exactly preserve mass across clusters. How realistic is this? Furthermore, does the j have to uniquely assigned to a k? Or can two source clusters push their mas to a target cluster? If not, this seems to immediately rule out DA between datasets where the target domain has fewer labels that source domain (which, fine, is not that common in classic DA, but is very common in large-scale image pretraining/fine-tuning). But beyond that,
* Is the bound computable? Given g, $f_N$, $p_N$, (C) and (A) seem straightforward to estimate from samples. What about (L)? The second distribution relies on $p_T$ and $p_T(Z|Y=k)$ which are not directly available in UDA.
* In related work, OT approaches for UDA: it is stated that Rakotomamonjy et al. (2020) solves GeTarS with OT, buth then two lines later the paper claims OSTAR is the first one to to this. Is the *mapping* the keyword here? If so, would be nice to empahsize that this is the main difference between this paper and Rakotomamonjy et al. (2020).

Typos:
* It seems that Asssumption 4 should read $p_T(Z| Y=i)$ (not k).
* Pg 8 conforts->supports/confirms?

**Summary Of The Paper:**

This paper proposes a method for unsupervised domain adaptation under conditional and label shift, one of the most challening versions of DA. At the core of this method is an optimal transport problem that seeks a mapping between source and target domains *and* a class proportion vector that minimize a total transportation cost between the domains. Based on this formulation, and a set of 4 sensible assumptions, the paper provides theoretical results in the form of unicity of the solution and classic generalization bound on the target domain risk. The implemented algorithm is inspired (though not identical) to this objective, which is extended with a few other terms: a constraint on the the label proportions based on a confusion matrix, a classification loss for the mapped source samples, and a relaxed version of the OT's pushforward constraint, and finally, two objectives that seek to improve the discriminativity of target representations. The paper then proceeds to validate the propose method in a wide array of benchmark DA datasets, with various levels of class unbalance, whereby it shows that the method repeatedly and significantly outperforms SOTA UDA alternatives.

**Summary Of The Review:**

This paper makes a solid contribution towards the challening problem of unsupervised domain adaptation under label and conditional shift. Based on a sound and (to the best of my knowledge) novel variant of the parametric OT formulation, the method proposed achieves remarkable results across a wide range of benchmark tasks. All things considered, I think this paper would be a good contribution to ICLR and would be relevant to the OT and DA communities.

---

> ### Author Response · Authors · 2021-11-18
> **Answers to Reviewer FHUc - Part 1/3**
>
> **Weaknesses**
>
> *How relevant are our theoretical results for our objective function ?* We highlight the relations between the terms in our upper-bound (2) and the terms in our objective function (CAL). Each term in (CAL) explicitly minimizes a term, (C), (A) and (L) in the upper-bound (2).
> * First,  $\min_{f_N} L_c^g(f_N, N)$ minimizes (C).
> * Second $\min_\phi L_{wd}^g(\phi, p_N^Y)$ s.t. $p_N^Y = argmin_{p \geq 0, p \in \Delta_K} \frac{1}{2}||\hat{p_T^Y}-\hat{\mathbf{C}} \dfrac{p}{p_S^Y}||_{2}^{2}$ minimizes term (A).
> * Third, our OT penalization $\lambda_{OT} L_{OT}^g(\phi)$ constrains alignment to solve our OT problem in (OT); under this constraint, Proposition 1 shows that term (L) equals to zero when term (A) equals to zero. For (L), since target labels are unknown, it is difficult to obtain formal guarantees outside the optimum.
>
> To summarize, the terms in the objective function minimize terms (C) and (A) in upper-bound (2) and at the optimum of problem (OT) (i.e. when marginals are aligned), term (L) vanishes. These results are valid for a fixed input representation obtained with a given encoder. Our finding is that improved discriminativity is obtained via semi-supervised learning, corresponding to terms (SS) and (SSg) in Section 4. (SSg) aims at reducing the value of the upper-bound (2) by changing target representations. Indeed, since $g$ has been trained with source labels, nothing guarantees that term (C) at optimum (when (A) and (L) are zero) is low. In conclusion, all terms in our objective function act on our upper-bound and allow training a classifier $f_N$ with controlled target risk.
>
> *Missing runtime / complexity analysis*
> In practise on USPS $\rightarrow$ MNIST, the runtimes in seconds on a NVIDIA Tesla V100 GPU machine are the following: DANN: 22.75s, $WD_{\beta=0}$: 59.25s, MARSc: 72.87s, MARSg: 2769.06s, IW-WD: 74.17s, OSTAR+IM: 89.72s. We observe that (i) computing Wasserstein distance ($WD_{\beta=0}$) is slower than approaching $\mathcal{H}$-divergence (DANN), (ii) runtimes for domain-invariant GeTarS baselines are slightly higher than for $WD_{\beta=0}$ as proportions are additionally estimated, (iii) domain-invariant GeTarS baselines have similar runtimes apart from MARSg which uses GMM, (iv) our model, OSTAR+IM, has slightly higher runtime than GeTarS baselines other than MARSg. We now provide some further analysis on the computational cost and memory in OSTAR. We denote $K$ the number of classes, $d$ the dimension of latent representations, $n$ and $m$ the number of source and target samples.
> * Memory: Proportion estimation is based on the method in Lipton et al. (2018) and requires storing the confusion matrix $\hat{\mathbf{C}}$ and predictions $\hat{p_T^Y}$ with a memory cost of $O(K^2)$. Encoding is performed by a deep NN $g$ into $\mathbb{R}^d$ and classification by a shallow classifier from $\mathbb{R}^d$ to $\mathbb{R}^K$. Alignment is performed with a ResNet $\phi:\mathbb{R}^d \rightarrow \mathbb{R}^d$ in the latent space which has an order magnitude less parameters than $g$. Globally, memory consumption is mostly defined by the number of parameters in the encoder $g$.
> * Computational cost comes from: (i) proportion estimation, which depends on $K, n + m$ and is solved once every 5 epochs with small runtime; (ii) alignment between source and target representations, which depends on $d, n + m$ and $N_v$ (the number of critic iterations). This step updates less parameters than domain-invariant methods which align with $g$ instead of $\phi$; this may lead to speedups for large encoders. The transport cost $L^g_{OT}$ depends on $n, d$ and adds small additional runtime; (iii) classification with $f_N$ using labelled source samples, which depends on $d,K,n$. In a second stage, we improve target discriminativity by updating the encoder $g$ with semi-supervised learning; this depends on $d,K,n+m$.

---

> > ### Author Response · Authors · 2021-11-18
> > **Answers to Reviewer FHUc - Part 2/3**
> >
> > *Intuition / motivation for our assumptions with a focus on assumption 2*
> > Our four assumptions are required for deriving our theoretical guarantees. All GeTarS papers that propose theoretical guarantees also rely on a set of assumptions, either explicit or implicit. Assumptions 1 and 4 are common to several papers. Assumption 1 is easily met in practice since it could be forced by training a classifier on the source labels. Assumption 4 is said to be met in high dimensions (Redko et al. (2019); Garg et al. (2020)). Assumption 3 says that source and target clusters from the same class are globally (there is a sum term in the condition) closer one another than clusters from two different classes. This assumption is required to cope with the absence of target labels. Because of the sum term, it could be considered as a reasonable hypothesis. It is milder than the hypotheses made in papers offering guarantees similar to ours (Zhang et al. (2013); Gong et al. (2016)). Assumption 2 is new to this paper. It states that $\phi$ maps a $j$ source conditional to a unique $k$ target conditional i.e. it guarantees that mass of a source conditional will be entirely transferred to a target conditional and will not be split across several target conditionals. This property is key to show that OSTAR matches source conditionals and their corresponding target conditionals, which is otherwise very difficult to guarantee under GeTarS as both target conditionals and proportions are unknown. As noted by the reviewer, this assumption has some restrictions, despite being milder than existing assumptions in GeTarS. First, mass from a source conditional might in practise be split across several target conditionals i.e. as the reviewer states it "$\phi$ might not preserve mass across clusters". In practise, our optimization problem in (CAL) mitigates this problem by learning how to reweight source conditionals to improve alignment. We could additionally apply Laplacian regularization (Ferradans et al. (2013), Carreira-Perpinan et al. (2014)) into our OT map $\phi$ but this was not developed in the paper. Second, it assumes a closed-set UDA problem i.e. label domains are the same $\mathcal{Y}_S=\mathcal{Y}_T$. As mentioned by the reviewer, the setting where $\mathcal{Y}_S \neq \mathcal{Y}_T$ is more realistic in large scale image pretraining and is another interesting follow-up. We refer to You et al. (2019) for further discussion on the various possible scenarios. Note that OSTAR could address empirically open-set UDA ($\mathcal{Y}_T \subset \mathcal{Y}_S$) by simply applying L1 regularization on $p_N^Y$ in our objective function (CAL). This forces sparsity and allows "loosing" mass when mapping a source conditional. It avoids the unwanted "negative transfer" setting described by the reviewer when "two source clusters push their mass to a same target cluster" i.e. when source clusters, whose labels are absent on the target, are aligned with some target clusters.
> >
> > *How to choose the encoder ?*
> > A common practise in UDA is to choose a deep encoder $g$ and a shallow classifier $f$ i.e. to perform adaptation on the deeper layers of the network (Long et al. (2015)). A rationale for this is to enforce the learned representations to be linearly separated in the latent space obtained with the encoder's last layer. A shallow classifier operates as a constraint on the encoder, by enforcing a clear class separation in the encoder space (e.g. linear if the classifier is linear). This allows in practise to better satisfy our assumptions. Throughout this work, we followed this intuition and did not try shallow encoders. How to select an appropriate encoder is an interesting question. Since we tackle an unsupervised problem, there is no principled answer. However, one may use the information provided by the upper-bound to the target risk in (2) to get some guidance. Yet, in practise, not all terms in our upper-bound are computable (see the answer "Is the upper-bound computable?" below). To avoid being too dependent on the initial encoder trained on the source data, we proposed an extension to our model based on semi-supervised learning (here IM) which updates the encoder to better enforce class separability on the target domain. We found in practise that this helped reduce the value of our initial upper-bound (to demonstrate this, we assumed knowledge of target conditionals to compute the value of the upper-bound). Table 1 below compares the values of terms (C), (A) and (L) from the upper-bound (2) with IM (columns OSTAR+IM) and without IM (columns OSTAR). Remember that term (C) is related to classification and terms (A) and (L) to alignment. Terms (A) and (L) are computed with the primal formulation of OT using POT https://pythonot.github.io. Term (C) is easily computed using labelled samples in domain $N$. We include this analysis and table in the revision (Table 6 in the revision).

---

> > > ### Author Response · Authors · 2021-11-18
> > > **Answers to Reviewer FHUc - Part 3/3**
> > >
> > > Table 1. Value after 50 training epochs of term (A) -$W_1(p_N^{\phi}(Z), p_T(Z))$- ($\downarrow$), term (L) -$W_1(\sum_{k=1}^{K} p_{N}^{Y=k} p_T(Z|Y=k), p_T(Z))$- ($\downarrow$) and term (C) -$\epsilon_N^g(f_N)$- ($\downarrow$) without and with IM in OSTAR. Best results are in **bold**.
> > >
> > > |          | Term | (A) | Term    | (L)            | Term  | (C)|
> > > |----------|----------|----------|------------------------|----------|----------|----------|
> > > | Setting  | OSTAR    | OSTAR+IM | OSTAR                  | OSTAR+IM | OSTAR    | OSTAR+IM |
> > > |          |          |          | MNIST|   $\rightarrow$       |    USPS       |          |
> > > | Balanced | $49.83$ | $\textbf{16.56}$ | $1.45$ | $\mathbf{0.18}$ | $2.19 \times 10^{-3}$ | $\mathbf{0.918 \times 10^{-3}}$|
> > > | Mild     | $39.31$ | $\textbf{19.13}$ | $10.78$ | $\mathbf{0.24}$ | $1.65 \times 10^{-3}$ | $\mathbf{0.931 \times 10^{-3}}$|
> > > | High    | $38.60$ | $\textbf{21.10}$ | $12.78$ | $\mathbf{0.74}$ | $1.76\times 10^{-3}$ | $\mathbf{0.510 \times 10^{-3}}$|
> > > |          |          |          | USPS |  $\rightarrow$ |    MNIST             |          |
> > > | Balanced  | $235.43$ | $\textbf{86.65}$ | $7.08$ | $\mathbf{0.52}$ | $4.46 \times 10^{-3}$ | $\mathbf{0.495 \times 10^{-3}}$|
> > > | Mild   | $188.67$ | $\textbf{104.66}$ | $20.99$ | $\mathbf{1.05}$ | $3.98 \times 10^{-3}$ | $\mathbf{0.399 \times 10^{-3}}$|
> > > | High  | $181.64$ | $\textbf{123.83}$ | $21.62$ | $\mathbf{0.82}$ | $4.51\times 10^{-3}$ | $\mathbf{0.616 \times 10^{-3}}$|
> > >
> > > **Questions / Comments / Suggestions**
> > >
> > > *Discrepancy between objective (OT) and the Monge OT problem (9)*
> > > (OT) is the extension of (9) to the setting where $p_S^Y \neq p_T^Y$ with $p_T^Y$ unknown. This extension aims at matching conditional distributions regardless of how their mass differ and learns a weighting term $p_N^Y$ for source conditional distributions which addresses label shift settings. When $p_S^Y = p_T^Y$, these two formulations are equivalent under Assumption 1 if we fix $p_N^Y=p_S^Y=p_T^Y$.
> > >
> > > *Is the upper-bound computable ?*
> > > In the upper-bound in (2), terms (A) and (C) are computable while term (L), $W_1\Big(p_T(Z), \sum_{k=1}^K p_{N}^{Y=k} p_T(Z|Y=k)\Big)$, is not as it requires knowing target conditional distributions. (L) could be estimated in contexts such as semi-supervised DA, but not in ours. Note that alternative bounds in the literature are not computable either (Combes et al. (2020), Shui et al. (2021), Rakotomamonjy et al. (2020), Ben-David et al. (2010)).
> > >
> > > *Main difference with Rakotomamonjy et al. (2020)*
> > > Rakotomamonjy et al. (2020) is a domain-invariant OT method for GeTarS which relies on solving two OT problems, one for label estimation and one for alignment while OSTAR solves a single OT problem at fixed representations, without constraining domain-invariance. OSTAR is then novel owing to the explicit mapping. We clarified this in the revision.
> > >
> > > **Typos**
> > >
> > > We thank the reviewer for pointing out these typos, which we have corrected in the revision.
> > >
> > > **References in this answer**
> > >
> > > Zachary C. Lipton, Yu-Xiang Wang, and Alexander J. Smola.  Detecting and correcting for labelshift with black box predictors. ICML 2018.
> > >
> > > Ievgen Redko, Nicolas Courty, Réemi Flamary, and Devis Tuia. Optimal transport for multi-source domain adaptation under target shift. AISTATS 2019.
> > >
> > > Saurabh Garg, Yifan Wu, Sivaraman Balakrishnan, and Zachary Lipton.   A unified view of labelshift estimation. NeurIPS 2020.
> > >
> > > Kun Zhang, Bernhard Schölkopf, Krikamol Muandet, and Zhikun Wang.  Domain adaptation under target and conditional  shift. ICML 2013
> > >
> > > Mingming  Gong,  Kun  Zhang,  Tongliang  Liu,  Dacheng  Tao,  Clark  Glymour, and  Bernhard Scholkopf. Domain  adaptation  with  conditional  transferable  components. ICML 2016.
> > >
> > > S. Ferradans, N. Papadakis, J. Rabin, G. Peyré, and J.-F. Aujol, “Regularized discrete optimal transport,” in Scale Space and Variational Methods in Computer Vision, SSVM, 2013, pp. 428–439.
> > >
> > > M. Carreira-Perpinan and W. Wang, “LASS: A simple assignment model with laplacian smoothing,” in AAAI Conference on Artificial Intelligence, 2014.
> > >
> > > Kaichao You, Mingsheng Long, Zhangjie Cao, Jianmin Wang, Michael I. Jordan. Universal Domain Adaptation. CVPR, 2019.
> > >
> > > Mingsheng Long, Yue Cao, Jianmin Wang, and Michael I. Jordan. 2015. Learning transferable features with deep adaptation networks. ICML 2015.
> > >
> > > Rémi Tachet des Combes, Han Zhao, Yu-Xiang Wang, and Geoff Gordon.  Domain adaptation with conditional distribution matching and generalized label shift. NeurIPS, 2020.
> > >
> > > Changjian Shui, Zijian Li, Jiaqi Li, Christian Gagnée, Charles X Ling, and Boyu Wang. Aggregating from multiple target-shifted sources. ICML 2021.
> > >
> > > Alain Rakotomamonjy, Rémi Flamary, Gilles Gasso, Mokhtar Z. Alaya, Maxime Berar, and Nicolas Courty. Match and reweight strategy for generalized target shift. ArXiv.
> > >
> > > Ben-David S., Blitzer J., Crammer K., Kulesza A., Pereira F., Vaughan J.W. A theory of learning from different domains. Machine Learning 79(1), 151–175 (2010)

---

### Author Response · Authors · 2021-11-18
**Revision and summary**

We thank the reviewers for their feedback on our paper "Mapping conditional distributions for domain adaptation under generalized target shift". We are encouraged they found our paper well written (R-FHUc, R-cdrt), our approach well motivated (R-FHUc, R-9ZXq) and our theoretical analysis to be a strength (R-FHUc, R-cdrt, R-9ZXq). We are glad they also acknowledge that our experimental results are thorough (R-FHUc) and state-of-the-art (R-FHUc, R-cdrt, R-9ZXq). We address comments and questions below. The main clarifications are the following:
* R-FHUc: We further explain the links between our objective function (CAL) + (SSg) and our upper-bound in (2). We also discuss the motivation behind our assumptions and especially the restrictions induced by assumption 2 as pointed out by R-FHUc. Finally, we discuss the role of the encoder and the difficulty in comparing two candidate encoders in an unsupervised setting. We circumvent this difficulty by improving our initial encoder via an extension based on Information Maximization (IM).
* R-cdrt: We compare OSTAR against related papers mentioned by R-cdrt. We further explain how our model improves target discriminativity with IM. We discuss the flexibility of Proposition 1. Finally, we extend our experimental ablation study on the effect of our transport cost $L_{OT}^g(\phi)$ in our objective function (CAL) with additional values of hyperparameter $\lambda_{OT}$ and analyze the edge case where $\lambda_{OT}$ is high.
* R-9ZXq: We clarify the usefulness of IM for OSTAR. With an additional experiment, we show that IM combined to alignment helps better optimize the three functions $\phi, g, f_N$ involved in our upper-bound in eq. (2) as it reduces in practise the value of our upper-bound. In the discussion, we provide more explanations and intuitions on why IM is connected to our theory and why it improves performance.

We incorporate the corresponding main corrections in blue in the revision.

---

### Decision · Program_Chairs · 2022-01-20

**Decision:**

Accept (Poster)

**Comment:**

This paper considers the generalized target shift setting for domain adaptation and proposes an optimal transport map-based approach to it. The considered setting for domain adaptation is rather general and of practical use. The proposed method seems sensible, as supported by the theoretical identifiability and empirical results.

It is worth noting that the way to cite previous work seems to be improved. For instance, in the first paragraph of Introduction, the authors reviewed various settings for domain adaptation. For model shift, the authors cited previous work. However, when discussing covariate shift, target shift, and generalized target shift, the authors did not cite the original work that provides the categorization. For completeness, the authors may want to consider including the setting of conditional shift as well, which has received a number of applications in domain adaptation in computer vision. I believe the categorization of target shift, conditional shift, and generalized target shift was provided by Zhang et al. (2013). This work should also be cited when the authors give the problem definition in Section 2.1. The quality of the paper will be even better if the authors cite previous work in all the right places--this may also make the authors' contribution clearer.